# DPMFormer: Dual-Path Mamba-Transformer for Efficient Image Super-Resolution

## Abstract

Vision Transformers have achieved outstanding performance in image super-resolution (SR), but existing lightweight models rely on window-based attention, limiting their ability to model global dependencies essential for high-quality reconstruction. To address these challenges, we present **DPMFormer**, a Dual-Path Mamba–Transformer architecture for lightweight image super-resolution. Rather than a simple combination of a state-space model and a Transformer, the design couples two streams throughout the network with deep dual fusions. On the Transformer side, an Enhanced Transformer Layer (ETL) replaces self-attention with Spatial–Channel Correlation (SCC) and a Depthwise-SwiGLU Feed-Forward Network (DW-SwiFFN). On the Mamba side, Lightweight Bidirectional Mamba Layers (LBi-ML) implement single-pass bidirectionality via channel split and sequence reversal with additive cross coupling. The streams interact at two levels: within each block, a Cross-Attention Layer (CAL) performs fixed, non-overlapping cross fusion, and across blocks, Inter-branch Exchange Bridges (IEB) use resolution-preserving $1 \times 1$ adapters around tokenization to align channel spaces in both directions. Besides, we employ RMSNorm to reduce normalization overhead and, under our setup, observe modest, configuration-dependent gains. Extensive experiments show that DPMFormer reduces MACs by 47.3G (21%) and parameters by 21K under $2\times$ upsampling compared to HiT-SR, while almost achieving state-of-the-art performance across five benchmarks. Measured on an RTX 4090, our method reaches 668 ms latency, yielding $1.59\times$ and $2.33\times$ speedups over MambaIR and CATANet, respectively. The code will be publicly released.

## 1 Introduction

Image super-resolution (SR) constitutes a fundamental yet inherently ill-posed inverse problem in computer vision, aiming to reconstruct high-resolution images from low-resolution inputs. With the rapid proliferation of mobile applications, real-time video streaming platforms, and edge computing devices, the development of computationally efficient models capable of delivering high-quality super-resolution results has become increasingly critical for practical deployment scenarios. Over the past decade, convolutional neural networks (CNNs) have dominated SR research (Dong et al., 2016a;b; Lim et al., 2017; Niu et al., 2020; Li et al., 2025), delivering impressive reconstruction quality. However, CNN-based architectures heavily rely on stacked convolutional layers, limiting each pixel's receptive field and hindering the modeling of global context, ultimately constraining further performance improvement.

Vision Transformers have rapidly advanced SR by capturing long-range dependencies via windowed self-attention (W-SA), as demonstrated in SwinIR (Liang et al., 2021), ELAN (Zhang et al., 2022a), SRFormer (Zhou et al., 2023), and HiT-SR (Zhang et al., 2025a). Directly modeling global context without windows incurs prohibitive computational costs, while windowing constrains the effective receptive field and thus limits model expressiveness—even shifted-window schemes only partially mitigate this trade-off. Nevertheless, capturing global context is essential for high-quality super-resolution, as it allows models to leverage distant spatial correlations and semantic coherence to generate structurally consistent and perceptually realistic high-resolution outputs.

To address these fundamental limitations, we propose Dual Path Mamba-Transformer (DPMFormer), motivated by Mamba's (Gu & Dao, 2023) ability to efficiently model long-range dependencies with linear computational complexity, unlike Transformers' quadratic complexity in global attention. Our approach leverages complementary strengths: Mamba excels at efficient global modeling, while windowed attention demonstrates superior local performance with lightweight computation. DPMFormer employs a dual-branch architecture to exploit complementary spatial modeling capabilities: the Transformer branch employs windowed attention for efficient local feature extraction, while the Mamba branch utilizes our lightweight bi-directional Mamba (LBi-Mamba) for linear-complexity global dependency modeling. To facilitate progressive information exchange between global and local representations from the two branches, we couple Transformer and Mamba at two levels, within blocks and across blocks, systematically integrating local detail preservation and global contextual coherence within a unified framework. We further incorporate two optimizations: (1) Root Mean Square Normalization (RMSNorm) to improve computational efficiency, and (2) DW-SwiFFN, which enhances SwiGLU with depthwise convolution to achieve effective global-local contextual modeling.

The primary contributions of this work are:

- We propose DPMFormer, a novel dual-branch architecture that combines window-based Transformer and Mamba blocks, exploiting their complementary advantages to extract richer feature representations under lightweight constraints.

- We introduce a novel dual information exchange mechanism with cross-attention integrated within Mamba blocks that facilitates dynamic feature exchange and fusion across branches.

- Extensive experiments demonstrate DPMFormer achieves leading SR performance with fewer parameters and MACs, establishing a new baseline for efficient SR models.

## 2 RELATED WORK

### 2.1 EFFICIENT SUPER-RESOLUTION

Initially, several methods explored lightweight SR models based on CNNs and achieved promising progress (Ahn et al., 2018; Du et al., 2022). However, both model depth and feature extraction capacity have proven to be particularly important for advancing the performance of lightweight SR models. This led to the emergence of information distillation approaches (Hui et al., 2018; 2019), which have subsequently undergone a series of improvements (Liu et al., 2020). Meanwhile, models based on various non-Transformer attention mechanisms have also yielded excellent results in the development of lightweight SR (Zhao et al., 2020; Behjati et al., 2023). In addition, methods based on lattice filter banks have provided new perspectives for lightweight SR (Luo et al., 2020). Nevertheless, while CNNs address local feature extraction through stacked convolutions, they remain less effective in modeling long-range dependencies.

### 2.2 TRANSFORMER-BASED SUPER-RESOLUTION

The self-attention (SA) mechanism was introduced to address the challenge of modeling long-range dependencies, which CNNs struggle with (Vaswani et al., 2017). In recent years, to better capture such dependencies in high-level vision tasks like recognition and segmentation, a variety of Transformer-based approaches have been proposed, including ViT (Kolesnikov et al., 2021), DETR (Carion et al., 2020), and Swin Transformer (Liu et al., 2021). Recently, the effectiveness of Transformers has also been explored in low-level vision tasks, such as denoising, deraining, and especially image super-resolution (Chen et al., 2021a; 2023). For super-resolution, SwinIR (Liang et al., 2021) introduced a shifted window-based self-attention (SW-SA) mechanism for image restoration, striking a balance between performance and efficiency. SRFormer (Zhou et al., 2023) introduced permuted self-attention (PSA) to enjoy large windows with minimal overhead, achieving high performance with reduced computational complexity by designing an efficient cross-scale attention module. More recently, HiT-SR (Zhang et al., 2025a) proposed expanding hierarchical windows plus a linear spatial-channel correlation module, achieving state-of-the-art SR with high efficiency. However, window-based Transformers are inherently constrained by their limited receptive fields, which restricts their ability to capture global contextual information and long-range dependencies.

### 2.3 LINEAR-BASED SUPER-RESOLUTION

Several purely linear-based SR approaches have been proposed. SESR (Bhardwaj et al., 2022) devised Collapsible Linear Blocks (CLB), using overparameterized linear transformations that can be folded at inference time. LAPAR (Li et al., 2020) formulates the LR to HR mapping as a pixel-adaptive regression over a predefined filter dictionary, achieving state-of-the-art performance with only a few thousand parameters and extremely low computational overhead. Mamba-based architectures leverage structured state-space models to capture both local and global dependencies in linear time, making them highly suitable for super-resolution (SR) tasks (Gu et al., 2021; Gu & Dao, 2023). Guo et al. first demonstrated this potential with MambaIR (Guo et al., 2025b), which integrates local enhancement and channel-attention modules into the vanilla Mamba backbone. On standard SR benchmarks, MambaIR (Guo et al., 2025b) outperforms SwinIR (Liang et al., 2021) while maintaining comparable FLOPs. Building on this, Xiao and Wang proposed MamEVSR (Xiao & Wang, 2025) for event-based video SR: they interleave adjacent-frame tokens via iMamba blocks for spatio-temporal fusion, and use Mamba blocks to aggregate event and frame modalities, achieving state-of-the-art video SR with linear complexity. These works establish a new paradigm for state-space-powered SR and motivate further exploration of Mamba variants in low-level vision. However, Mamba is limited in super-resolution tasks due to its lack of spatial inductive bias and sensitivity to local textures, which hinders the reconstruction of fine-grained high-frequency details.

## 3 METHOD

We first outline the core methodology of DPMFormer, followed by detailed presentations of its block-level and layer-level designs, respectively.

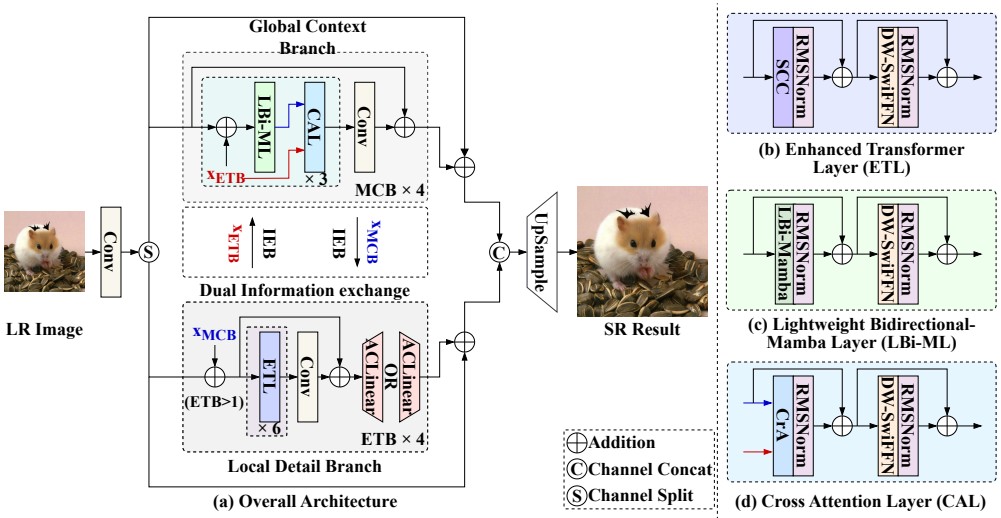

Figure 1: Overview of our DPMFormer architecture and its key components. "OR" indicates different forms of the Adjustment Channel Linear (ACLinear), where the preceding and succeeding ACLinear correspond to cases with increasing and decreasing channel dimensions, respectively. LBi-ML denotes the full layer and LBi-Mamba refers only to the state-space core.

### 3.1 OVERVIEW ARCHITECTURE

Transformer-based SR captures long-range dependencies, but the quadratic cost of dense attention typically forces windowing, which limits global context and hinders multi-scale aggregation. State-space models (SSMs) such as Mamba propagate long-range signals with favorable scaling yet lack content-adaptive weighting and often underperform on fine detail.

We seek a lightweight SR architecture that preserves the favorable scaling of SSMs, introduces content-adaptive fusion where it matters, and exchanges information between branches inside each block rather than only at stage boundaries. Therefore, Our Dual-Path Mamba–Transformer follows the general idea of combining an attention path with an SSM path, but it differs in three concrete, implementation-backed aspects, and Figure 1 annotates the two branches and their block interaction.

1. **Per-block cross fusion.** Each block applies a windowed Cross-Attention Layer (CAL), queries from the Transformer branch attend to keys, values from the Mamba branch, enabling continuous cross-branch exchange within every block.

2. **Single-pass bidirectional Mamba.** Rather than two forward/backward SSM scans, we form a bidirectional sequence by concatenating forward segments with their reversed counterparts and perform a single selective scan, retaining context while lowering overhead.

3. **Dual-path fusion and lightweight bridges.** On the attention side, we adopt a CrossFormer-style dual path with learnable fusion to coordinate complementary sub-streams (Wang et al., 2024). To keep fusion stable and low-overhead, we introduce lightweight Inter-branch Exchange Bridges (IEB) that shuttle features between branches.

## 3.2 Enhanced Transformer Block (ETB)

Each ETB consists of an Enhanced Transformer Layer (ETL) and a lightweight Adjustment Channel Linear (ACLinear) to replace the traditional Transformer block. Below we detail ETL and ACLinear.

**Enhanced Transformer Layer (ETL).** ETL comprises a Spatial–Channel Correlation (SCC), a DW-SwiFFN, and two RMSNorms. All ETLs are performed on non-overlapping windows with base size $w_{\text{base}}$ and expansion ratio $r$. The effective window and its token count are

$$(w_H, w_W) = (r\, w_{\text{base}},\ r\, w_{\text{base}}),\ m = w_H\, w_W. \tag{1}$$

Given a feature map with $L = H \cdot W$ tokens, ETL partitions tokens into $\frac{L}{m}$ windows.

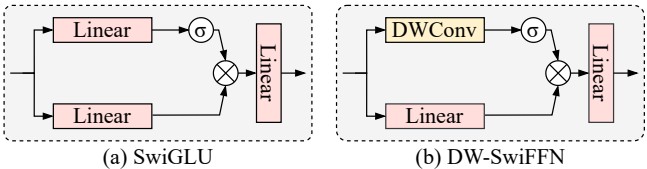

(a) SwiGLU     (b) DW-SwiFFN

Figure 2: Comparison between SwiGLU and DW-SwiFFN.

**Spatial–Channel Correlation (SCC).** Given a window feature tensor $\mathbf{X} \in \mathbb{R}^{m \times C}$ with $m = w_f^2$ tokens and $C$ channels, we first apply a head-wise linear projection and reshape it into $\mathbf{Q}, \mathbf{V} \in \mathbb{R}^{h \times m \times d_h}$, where $h$ is the number of heads and $C = h\, d_h$. We denote by $\widehat{\mathbf{V}} \in \mathbb{R}^{h \times m \times d_h}$ a lightweight value projection of $\mathbf{V}$, and by $\mathbf{B}_{rel} \in \mathbb{R}^{h \times m \times m}$ a learned relative positional bias. All matrix multiplications below are performed *independently for each head*: for head index $k$, we have $\mathbf{Q}^{(k)}, \mathbf{V}^{(k)}, \widehat{\mathbf{V}}^{(k)} \in \mathbb{R}^{m \times d_h}$ and $\mathbf{B}_{rel}^{(k)} \in \mathbb{R}^{m \times m}$. The spatial self-correlation (S–SC) and channel self-correlation (C–SC) are defined as

$$\text{S-SC}(\mathbf{Q}, \mathbf{V}) = \left(\frac{\mathbf{Q}\,\widehat{\mathbf{V}}^{\top}}{\alpha} + \mathbf{B}_{rel}\right)\widehat{\mathbf{V}}, \qquad \text{C-SC}(\mathbf{Q}, \mathbf{V}) = \left(\frac{\mathbf{Q}^{\top}\mathbf{V}}{m}\right)\mathbf{V}^{\top}, \tag{2}$$

where $\alpha > 0$ is a learned per-head scaling factor. The outputs of both terms have shape $\mathbb{R}^{h \times m \times d_h}$ and are concatenated along the channel dimension, then reshaped back to $\mathbb{R}^{m \times C}$ via a linear projection before the window merge.

DW-SwiFFN. Unlike the standard Transformer FFN, ETL adopts a depthwise SwiGLU-style feed-forward. Given an input window tensor $\mathbf{X} \in \mathbb{R}^{m \times C}$, we first project it to a higher-dimensional hidden space $\mathbf{U} = \text{Linear}_{up}(\mathbf{X}) \in \mathbb{R}^{m \times C_{ff}}$, apply a depthwise convolution $\text{DWConv}(\mathbf{X}) \in \mathbb{R}^{m \times C_{ff}}$,

and then use a gated nonlinearity:

$$\text{FFN}(\mathbf{X}) = \text{Linear}_{out}\big(\text{Linear}_{up}(\mathbf{X}) \odot \sigma\big(\text{DWConv}(\mathbf{X})\big)\big), \tag{3}$$

where $\sigma(\cdot)$ is the sigmoid, $\odot$ denotes element-wise (Hadamard) product, and $\text{Linear}_{out}$ is $\mathbb{R}^{m \times C_{ff}} \to \mathbb{R}^{m \times C}$ maps the hidden features back to the original channel dimension.

**RMSNorm.** All ETL submodules use RMSNorm for channel-wise normalization. Given $\mathbf{X} \in \mathbb{R}^{m \times C}$, RMSNorm computes $\text{RMSNorm}(\mathbf{X}) = \gamma \cdot \mathbf{X}/\sqrt{\frac{1}{C}\sum_c \mathbf{X}_{:,c}^2 + \epsilon}$, with learnable scale $\gamma \in \mathbb{R}^C$ and a small constant $\epsilon$. No mean subtraction or variance estimation is required.

**Adjustment Channel Linear (ACLinear).** ACLinear is a post-ETL, per-position linear projection that modulates channel width. Given $\mathbf{Z} \in \mathbb{R}^{m \times C_{in}}$, ACLinear applies $\text{ACLinear}(\mathbf{Z}) = \mathbf{Z}\,\mathbf{W}_{ac}^\top + \mathbf{b}_{ac}$, where $\mathbf{W}_{ac} \in \mathbb{R}^{C_{out} \times C_{in}}$ and $\mathbf{b}_{ac} \in \mathbb{R}^{C_{out}}$. Its mode alternates across successive ETBs with $C_1 < C_2$: expansion ($C_1 \to C_2$) redistributes features in a higher-dimensional subspace to enrich subsequent blocks, whereas compression ($C_2 \to C_1$) acts as a learned bottleneck that aggregates responses back to $C_1$.

### 3.3 MAMBA CROSS BLOCK (MCB)

A MCB stacks $d$ lightweight layers that alternate between an LBi-ML and a CAL. Let $\mathbf{X}_t \in \mathbb{R}^{L \times C}$ denote the Mamba-branch tokens and $\mathbf{Y}_t \in \mathbb{R}^{L \times C}$ the Transformer-branch tokens produced by the preceding ETB, where $L = HW$ is the number of tokens and $C$ the channel dimension. For layer index $t = 1, \ldots, d$,

$$\mathbf{X}_{t+1} = \begin{cases} \mathbf{X}_t + \text{Conv}\big(\text{CAL}(\mathbf{X}_t; \mathbf{Y}_t)\big), & t \text{ is even,} \\ \mathbf{X}_t + \text{Conv}\big(\text{LBi-ML}(\mathbf{X}_t + \mathbf{Y}_t)\big), & t \text{ is odd,} \end{cases} \tag{4}$$

where $\text{Conv} : \mathbb{R}^{L \times C} \to \mathbb{R}^{L \times C}$ is a $1 \times 1$ convolution applied in token space. This scheme injects cross-branch context at complementary sites in the attention and state-space branches.

**Cross-Attention Layer (CAL).** Within each block, CAL injects cross-branch context while maintaining resolution-independent attention by constraining computation to windows. We partition $\mathbf{X}_t$ and $\mathbf{Y}_t$ into non-overlapping windows of size

$$(w_H, w_W) = (w_f, w_f), \qquad m = w_H w_W = w_f^2. \tag{5}$$

For a window index $w$, let $\mathbf{X}_w, \mathbf{Y}_w \in \mathbb{R}^{m \times C}$ denote the Mamba and Transformer branch tokens in that window. We project and reshape them into multi-head queries, keys and values $\mathbf{Q}_{Y_w}, \mathbf{K}_{X_w}, \mathbf{V}_{X_w} \in \mathbb{R}^{h \times m \times d_h}$ with $C = h\,d_h$, and use a head-wise relative positional bias $\mathbf{B}_{rel} \in \mathbb{R}^{h \times m \times m}$. The per-window cross-attention update is

$$\text{CAL}_w(\mathbf{X}_w; \mathbf{Y}_w) = \text{softmax}\Big(\frac{\mathbf{Q}_{Y_w}\mathbf{K}_{X_w}^\top}{\sqrt{d_h}} + \mathbf{B}_{rel}\Big)\mathbf{V}_{X_w}, \tag{6}$$

which outputs a tensor in $\mathbb{R}^{h \times m \times d_h}$ that is then reshaped back to $\mathbb{R}^{m \times C}$ and merged over all windows.

**Lightweight Bi-directional Mamba Layer (LBi-ML).** LBi-ML enables linear-time long-range propagation by coupling the two branches within the SSM dynamics, and instantiates single-pass bidirectionality through channel splitting and sequence reversal, which reduces the scan constant vs. performing two scans; an additive cross-branch input is injected before the scan. After an input projection, channels are split into two halves, one half is reversed along the sequence axis to synthesize a forward-backward pair, then concatenated and fed to a single selective scan

$$\mathbf{X}_{ssm} = \text{Scan}(\text{DWConv}(\text{Concat}(\mathbf{H}_1 \to, \text{rev}(\mathbf{H}_2 \to)), \theta). \qquad \mathbf{H}_1, \mathbf{H}_2 = \widetilde{\mathbf{X}}_{InP}.\text{Split}. \tag{7}$$

where $\text{rev}(\cdot)$ reverses the sequence order, $\text{Scan}(\cdot; \theta)$ is a single selective state-space scan which shown as Figure 3, and DWConv is a depthwise 1D convolution preceding the scan.

### 3.4 Inter-branch Exchange Bridges (IEB)

We introduce the Inter-branch Exchange Bridges (IEB) to strengthen cross-branch interaction. IEB aligns Transformer and Mamba features via a three-step adapter, consists a patch unembedding, a ConvBlockD, and a patch embedding, where ConvBlockD comprises two $1 \times 1$ convolutions with an intermediate GELU, providing lightweight local refinement. It applied bidirectionally , from Transformer to Mamba and from Mamba to Transformer, yielding mutually reinforcing integration across receptive fields.

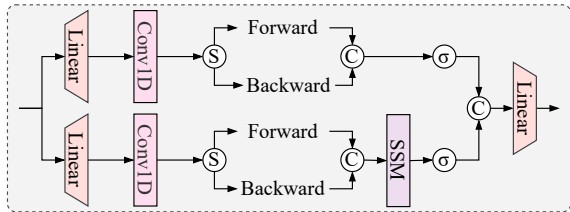

Figure 3: Detailed structure of the proposed LBi-Mamba.

## 4 Experiments and Analysis

### 4.1 Implementation Details

As in most lightweight architectures, we set the number of ETB and MCB blocks to 4. Each block stacks 6 layers: the ETB contains 6 ETL layers, while in the MCB we interleave 3 Mamba layers and 3 Cross Attention layers. We use $8 \times 8$ as the fixed window in CAL, $8 \times 8$ as the base window in ETL and realize multi-scale context by applying expand ratios r $\in$ [0.5,1,2,4,6,8], which map to effective window sizes [4,8,16,32,48,64] on $64 \times 64$ training patches. For the Transformer branch, the input and output channel widths of the four ETB blocks are [36,48,36,48] and [48,36,48,36], respectively. The Cross Mamba branch uses a constant width of 28.

All models are implemented in PyTorch and trained on $64 \times 64$ patches with a batch size of 16 for 500,000 iterations. We optimize using the $\ell_1$ loss and the AdamW (Loshchilov & Hutter, 2019) optimizer with parameters $\beta_1 = 0.9$, $\beta_2 = 0.99$, and weight decay $10^{-2}$. The initial learning rate is set to $5 \times 10^{-4}$ and is halved at iterations 250k, 400k, 450k, and 475k. During training, data augmentation includes random rotations ($90°$, $180°$, and $270°$), and horizontal flips. All ablations use the same seeds, data augmentations, and training schedule, results are averaged 5 runs.

### 4.2 Data and Evaluation

Following prior practice, we train on the DIV2K dataset (Agustsson & Timofte, 2017) and evaluate on five standard benchmarks: Set5 (Bevilacqua et al., 2012), Set14 (Zeyde et al., 2010), B100 (Martin et al., 2001), Urban100 (Huang et al., 2015), and Manga109 (Matsui et al., 2017). Low-resolution images are generated by bicubic downsampling of the high-resolution counterparts. We perform experiments under scale factors of $\times 2$, $\times 3$, and $\times 4$, and report peak signal-to-noise ratio (PSNR) and structural similarity index (SSIM) (Wang et al., 2004) measured on the Y channel of the YCbCr color space.

### 4.3 Comparison with State-of-the-Art Methods

We evaluate the proposed DPMFormer by comparing it against state-of-the-art efficient SR methods. The CNN-based algorithms include EDSR-B (Lim et al., 2017), CARN (Ahn et al., 2018), IMDN (Hui et al., 2019), LatticeNet (Luo et al., 2020), SRPN-Lite (Zhang et al., 2022b), GASSL-B (Wang et al., 2023b), and SRConvNet (Li et al., 2025). Transformer-based methods encompass HNCT (Fang et al., 2022), ELAN-L (Zhang et al., 2022a), Omni-SR (Wang et al., 2023a), SwinIR-NG (Choi et al., 2023), SRFormer-L (Zhou et al., 2023), HiT-SR (Zhang et al., 2025a), CATANet (Liu

Table 1: Quantitative comparison with state-of-the-art SR methods.The output size is set to $720 \times 1280$ for all scales to compute parameters and MACs. Best results are bold.

| Method | Scale | Complexity | | Set5 | Set14 | B100 | Urban100 | Manga109 |
|---|---|---|---|---|---|---|---|---|
| | | Params | MACs | PSNR/SSIM | PSNR/SSIM | PSNR/SSIM | PSNR/SSIM | PSNR/SSIM |
| EDSR-B (Lim et al., 2017) | ×2 | 1370K | 316.3G | 37.99/0.9604 | 33.57/0.9175 | 32.16/0.8994 | 31.98/0.9272 | 38.54/0.9769 |
| CARN (Ahn et al., 2018) | ×2 | 1592K | 222.8G | 37.76/0.9590 | 33.52/0.9166 | 32.09/0.8978 | 31.92/0.9256 | 38.36/0.9765 |
| IMDN (Hui et al., 2019) | ×2 | 694K | 158.8G | 38.00/0.9605 | 33.63/0.9177 | 32.17/0.9283 | | 38.88/0.9774 |
| LatticeNet (Luo et al., 2020) | ×2 | 756K | 169.5G | 38.06/0.9607 | 33.70/0.9187 | 32.20/0.8999 | 32.25/0.9288 | 38.94/0.9774 |
| SRPN-Lite (Zhang et al., 2022b) | ×2 | 609K | 139.9G | 38.10/0.9610 | 33.70/0.9189 | 32.25/0.9005 | 32.26/0.9294 | - |
| GASSL-B (Wang et al., 2023b) | ×2 | 689K | 158.2G | 38.08/0.9607 | 33.75/0.9194 | 32.24/0.9005 | 32.29/0.9298 | 38.92/0.9777 |
| SRConvNet (Li et al., 2025) | ×2 | 387K | 74.0G | 38.00/0.9605 | 33.58/0.9186 | 32.16/0.8995 | 32.05/0.9272 | 38.87/0.9774 |
| HNCT (Fang et al., 2022) | ×2 | 357K | 82.4G | 38.08/0.9608 | 33.65/0.9182 | 32.22/0.9001 | 32.22/0.9294 | 38.87/0.9774 |
| ELAN-L (Zhang et al., 2022a) | ×2 | 621K | 201.3G | 38.17/0.9611 | 33.94/0.9207 | 32.32/0.9009 | 32.76/0.9340 | 39.11/0.9782 |
| Omni-SR (Wang et al., 2023a) | ×2 | 772K | 194.5G | 38.22/0.9613 | 33.98/0.9210 | 32.36/0.9020 | 33.05/0.9363 | 39.28/0.9784 |
| SwinIR-NG (Choi et al., 2023) | ×2 | 1181K | 274.1G | 38.16/0.9613 | 33.94/0.9205 | 32.31/0.9013 | 32.78/0.9340 | 39.20/0.9781 |
| SRFormer-L (Zhou et al., 2023) | ×2 | 853K | 236.2G | 38.23/0.9613 | 33.94/0.9209 | 32.36/0.9019 | 32.91/0.9353 | 39.28/0.9785 |
| CATANet (Liu et al., 2025b) | ×2 | 477K | - | 38.28/0.9617 | 33.99/ 0.9217 | 32.37/0.9023 | 33.09/0.9372 | 39.37/0.9784 |
| HiT-SRF (Zhang et al., 2025a) | ×2 | 847K | 226.5G | 38.26/0.9615 | 34.01/0.9214 | 32.37/0.9023 | 33.13/0.9372 | 39.47/0.9787 |
| MambaIR-L (Guo et al., 2025b) | ×2 | 905K | 167.1G | 38.13/0.9610 | 33.95/0.9208 | 32.31/0.9013 | 32.85/0.9349 | 39.20/0.9782 |
| Hi-Mamba-T (Qiao et al., 2024) | ×2 | 870K | 178.0G | 38.24/0.9613 | 34.06/0.9215 | 32.35/0.9019 | 33.04/0.9358 | 39.28/0.9785 |
| SRMamba-T-S (Liu et al., 2025a) | ×2 | 653K | 121.5G | 38.25/0.9614 | **34.24/0.9225** | 32.35/0.9017 | 33.20/0.9366 | 39.53/0.9790 |
| MambaIRv2-L (Guo et al., 2025a) | ×2 | 774K | 286.3G | 38.26/0.9615 | 34.09/0.9213 | 32.36/0.9019 | 33.26/0.9378 | 39.35/0.9785 |
| **DPMFormer(Ours)** | ×2 | 826K | 179.2G | **38.29/0.9618** | 34.12/0.9219 | **32.40/0.9027** | 33.29/0.9382 | 39.54/0.9792 |
| EDSR-B (Lim et al., 2017) | ×3 | 1555K | 160.2G | 34.37/0.9270 | 30.28/0.8417 | 29.09/0.8052 | 28.15/0.8527 | 33.45/0.9439 |
| CARN (Ahn et al., 2018) | ×3 | 1592K | 118.8G | 34.29/0.9255 | 30.29/0.8407 | 29.04/0.8034 | 28.06/0.8493 | 33.50/0.9440 |
| IMDN (Hui et al., 2019) | ×3 | 703K | 71.5G | 34.36/0.9270 | 30.32/0.8417 | 29.09/0.8046 | 28.17/0.8519 | 33.61/0.9445 |
| LatticeNet (Luo et al., 2020) | ×3 | 765K | 76.3G | 34.40/0.9272 | 30.38/0.8426 | 29.15/0.8061 | 28.19/0.8528 | 33.63/0.9442 |
| SRPN-Lite (Zhang et al., 2022b) | ×3 | 615K | 62.7G | 34.47/0.9276 | 30.38/0.8425 | 29.16/0.8063 | 28.22/0.8534 | - |
| GASSL-B (Wang et al., 2023b) | ×3 | 691K | 70.4G | 34.47/0.9278 | 30.39/0.8430 | 29.15/0.8063 | 28.27/0.8546 | 33.77/0.9455 |
| SRConvNet (Li et al., 2025) | ×3 | 387K | 33.0G | 34.40/0.9272 | 30.30/0.8416 | 29.07/0.8047 | 28.04/0.8500 | 33.56/0.9443 |
| HNCT (Fang et al., 2022) | ×3 | 363K | 37.8G | 34.47/0.9275 | 30.44/0.8439 | 29.15/0.8063 | 28.28/0.8557 | 33.81/0.9459 |
| ELAN-L (Zhang et al., 2022a) | ×3 | 629K | 89.5G | 34.61/0.9288 | 30.55/0.8439 | 29.21/0.8073 | 28.69/0.8614 | 33.80/0.9478 |
| Omni-SR (Wang et al., 2023a) | ×3 | 780K | 88.4G | 34.70/0.9294 | 30.57/0.8469 | 29.28/0.8094 | 28.84/0.8656 | 34.22/0.9487 |
| SwinIR-NG (Choi et al., 2023) | ×3 | 1201K | 114.1G | 34.64/0.9293 | 30.58/0.8471 | 29.23/0.8093 | 28.75/0.8639 | 34.22/0.9488 |
| SRFormer-L (Zhou et al., 2023) | ×3 | 861K | 132.9G | 34.67/0.9296 | 30.57/0.8469 | 29.26/0.8099 | 28.81/0.8655 | 34.19/0.9489 |
| CATANet (Liu et al., 2025b) | ×3 | 550K | - | 34.75/0.9300 | 30.67/0.8481 | 29.28/0.8101 | 29.04/0.8689 | 34.40/0.9499 |
| HiT-SRF (Zhang et al., 2025a) | ×3 | 855K | 101.6G | 34.75/0.9300 | 30.61/0.8475 | 29.29/0.8106 | 28.99/0.8687 | 34.53/0.9502 |
| MambaIR-L (Guo et al., 2025b) | ×3 | 913K | 74.5G | 34.63/0.9288 | 30.54/0.8459 | 29.23/0.8084 | 28.70/0.8631 | 34.12/0.9479 |
| Hi-Mamba-T (Qiao et al., 2024) | ×3 | 880K | 80.0G | 34.76/0.9298 | 30.61/0.8472 | 29.27/0.8091 | 29.05/0.8693 | 34.42/0.9499 |
| SRMamba-T-S (Liu et al., 2025a) | ×3 | 661K | 55.7G | 34.73/0.9296 | 30.65/0.8475 | 29.32/0.8098 | **29.14**/0.8694 | 34.55/0.9500 |
| MambaIRv2-L (Guo et al., 2025a) | ×3 | 781K | 126.7G | 34.71/0.9298 | 30.68/0.8483 | 29.26/0.8098 | 29.01/0.8689 | 34.41/0.9497 |
| **DPMFormer(Ours)** | ×3 | 834K | 80.5G | **34.78/0.9302** | 30.69/0.8486 | **29.34/0.8110** | 29.12/**0.8699** | 34.58/0.9506 |
| EDSR-B (Lim et al., 2017) | ×4 | 1518K | 114.0G | 32.09/0.8938 | 28.58/0.7813 | 27.57/0.7357 | 26.04/0.7849 | 30.35/0.9067 |
| CARN (Ahn et al., 2018) | ×4 | 1592K | 90.9G | 32.13/0.8937 | 28.60/0.7806 | 27.59/0.7349 | 26.07/0.7837 | 30.47/0.9084 |
| IMDN (Hui et al., 2019) | ×4 | 715K | 40.9G | 32.21/0.8948 | 28.58/0.7811 | 27.56/0.7353 | 26.04/0.7838 | 30.45/0.9075 |
| LatticeNet (Luo et al., 2020) | ×4 | 777K | 43.6G | 32.18/0.8943 | 28.61/0.7812 | 27.57/0.7355 | 26.14/0.7784 | 30.54/0.9075 |
| SRPN-Lite (Zhang et al., 2022b) | ×4 | 623K | 35.8G | 32.24/0.8958 | 28.68/0.7836 | 27.63/0.7373 | 26.16/0.7875 | - |
| GASSL-B (Wang et al., 2023b) | ×4 | 694K | 39.9G | 32.17/0.8950 | 28.66/0.7835 | 27.62/0.7373 | 26.36/0.7888 | 30.70/0.9100 |
| SRConvNet (Li et al., 2025) | ×4 | 382K | 22.0G | 32.18/0.8951 | 28.61/0.7818 | 27.57/0.7359 | 26.06/0.7845 | 30.35/0.9075 |
| HNCT (Fang et al., 2022) | ×4 | 373K | 22.0G | 32.31/0.8957 | 28.71/0.7834 | 27.63/0.7381 | 26.20/0.7896 | 30.70/0.9112 |
| ELAN-L (Zhang et al., 2022a) | ×4 | 640K | 53.7G | 32.43/0.8975 | 28.78/0.7848 | 27.69/0.7406 | 26.54/0.7982 | 30.92/0.9150 |
| Omni-SR (Wang et al., 2023a) | ×4 | 792K | 50.9G | 32.49/0.8988 | 28.78/0.7859 | 27.71/0.7415 | 26.64/0.8018 | 31.02/0.9151 |
| SwinIR-NG (Choi et al., 2023) | ×4 | 1201K | 64.4G | 32.44/0.8980 | 28.83/0.7870 | 27.73/0.7422 | 26.67/0.8032 | 31.09/0.9161 |
| SRFormer-L (Zhou et al., 2023) | ×4 | 873K | 62.8G | 32.51/0.8988 | 28.82/0.7872 | 27.73/0.7422 | 26.68/0.8032 | 31.17/0.9156 |
| CATANet (Liu et al., 2025b) | ×4 | 535K | 46.8G | 32.58/0.8998 | 28.90/0.7880 | 27.75/0.7427 | 26.87/0.8081 | 31.31/0.9183 |
| HiT-SRF (Zhang et al., 2025a) | ×4 | 866K | 58.0G | 32.55/0.8999 | 28.87/0.7880 | 27.75/0.7432 | 26.80/0.8069 | 31.26/0.9171 |
| MambaIR-L (Guo et al., 2025b) | ×4 | 924K | 42.3G | 32.42/0.8977 | 28.74/0.7847 | 27.68/0.7400 | 26.52/0.7983 | 30.94/0.9135 |
| Hi-Mamba-T (Qiao et al., 2024) | ×4 | 890K | 45.0G | 32.52/0.8995 | 28.80/0.7873 | 27.75/0.7429 | 26.81/0.8072 | 31.35/0.9186 |
| SRMamba-T-S (Liu et al., 2025a) | ×4 | 672K | 32.8G | 32.55/0.8991 | 28.85/0.7876 | 27.76/0.7423 | 26.78/0.8052 | 31.32/0.9184 |
| MambaIRv2-L (Guo et al., 2025a) | ×4 | 790K | 75.6G | 32.51/0.8992 | 28.84/0.7878 | 27.75/0.7426 | 26.82/0.8079 | 31.24/0.9182 |
| **DPMFormer(Ours)** | ×4 | 845K | 56.9G | **32.60/0.9001** | 28.91/0.7882 | 27.77/0.7436 | 26.93/0.8086 | 31.40/0.9189 |

et al., 2025b). Finally, Mamba-based approaches MambaIR-L (Guo et al., 2025b), Hi-Mamba-T (Qiao et al., 2024), SRMamba-T-S (Liu et al., 2025a) and MambaIRv2-L (Guo et al., 2025a).

**Quantitative Comparison.** Table 1 summarizes PSNR/SSIM, parameters, and MACs against leading lightweight SR models. At $2\times$, DPMFormer uses 826K parameters and 179.2G MACs, lower than SRFormer-Light and HiT-SR, while delivering leading or competitive results across the benchmarks; on Urban100 it exceeds HiT-SRF by 0.06 dB and MambaIR by 0.34 dB. Table 2 shows that at $4\times$ on an RTX 4090, DPMFormer runs in 668 ms and 37%/57% faster than MambaIR/CATANet while maintaining superior fidelity, achieving strong quality–efficiency trade-offs than HiT-SR.

**Qualitative Comparisons.** Figure 4 presents qualitative comparisons on challenging SR cases. Methods biased toward local cues with limited receptive fields often blur details and introduce artifacts (e.g. "img_085"); leveraging dual-branch modeling, DPMFormer strengthens long-range de-

Table 2: Comparisons between SOTA efficient SR models and our approaches. Running time are measured under $2\times$ upscaling on RTX 4090 GPU with the output image size set to $720 \times 1280$.

| Method | SwinIR | SRFormer | HiT-SRF | CATANet | MambaIR | DPMFormer(Ours) |
|---|---|---|---|---|---|---|
| running time (ms) | 3723 | 3819 | **516** | 1552 | 1063 | 668 |

pendencies to yield sharper textures and finer detail. In geometry-heavy scenes (e.g. "img_076") and text regions "EienNoWith", it better preserves global structure and produces clearer glyph contours (e.g., R, E), recovering both local detail and overall layout.

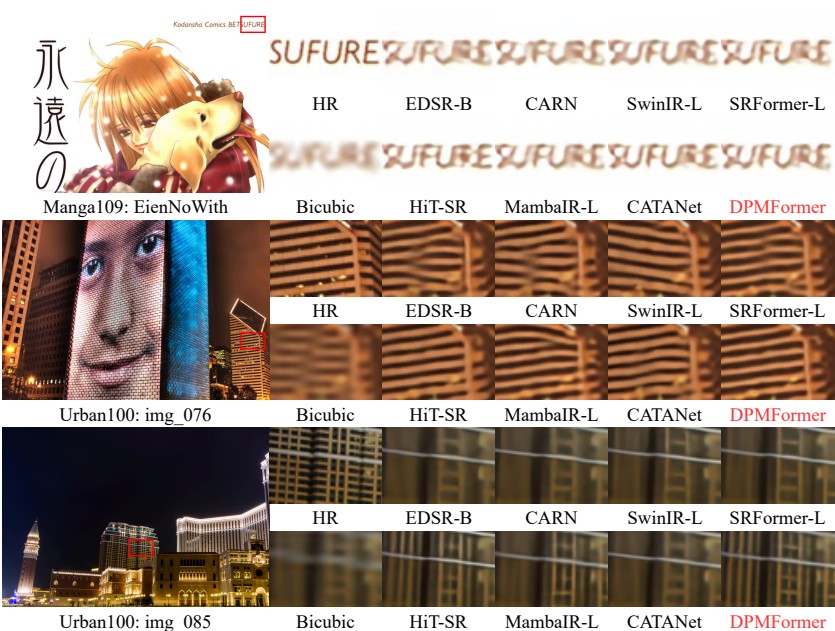

Figure 4: Qualitative comparisons for image SR ($4\times$) in challenging scenes. Red denotes ours.

## 4.4 ABLATION STUDY

We conduct an ablation study on the DPMFormer backbone to evaluate the effectiveness of each proposed component. All model variants were trained and tested using the aforementioned experimental settings and datasets on $2\times$ upsampling factor.

**Dual-Branch Architecture.** To verify the effect of Within our dual-branch framework, we instantiate a baseline (HiT-V) by configuring the HiT-SRF backbone with stage-wise input/output are [36, 48, 36, 48]/[48, 36, 48, 36]. Appending a 28-channel Mamba branch yields Mamba-Transformer (MAT). As reported in Table 3, MAT consistently improves reconstruction.

**CAL.** To validate the effectiveness of incorporating cross-attention into the Mamba branch, we replace each standard Mamba layer with a Cross-Attention Layer (CAL), yielding the baseline Dual-Path Mamba–Transformer model (DPMFormer-B). As shown in Table 3, the addition of cross-attention further enhances the model's reconstruction performance on complex textures.

**Dual-Branch Channel Widths.** We investigate the impact of channel widths for the ETB and MCB by assigning various configurations to each branch to identify the optimal setup. The results, presented in Table 3, demonstrate that using the VAR configuration for the ETB and a channel width of 28 for the MCB achieves the best performance.

Table 3: Dual-branch architecture and cross-attention effectiveness, where CT denotes the channel width of the Transformer branch and CM denotes the channel width of the Mamba branch.

| Method | CT | CM | Set5 | Set14 | B100 | Urban100 | Manga109 |
|---|---|---|---|---|---|---|---|
| HiT-V | VAR | - | 38.15/0.9615 | 33.88/0.9205 | 32.30/0.9012 | 32.69/0.9335 | 39.11/0.9782 |
| MAT | VAR | 28 | 38.16/0.9615 | 33.87/0.9198 | 32.30/0.9011 | 32.91/0.9349 | 39.22/0.9781 |
| DPMFormer-B | VAR | 28 | **38.18/0.9617** | **33.89/0.9206** | **32.32/0.9013** | **32.97/0.9356** | **39.28/0.9785** |
| | 48 | 12 | 38.26/0.9617 | 34.02/0.9213 | 32.38/0.9026 | 33.15/0.9373 | 39.49/0.9787 |
| DPMFormer | VAR | 28 | **38.29/0.9618** | **34.12/0.9219** | **32.40/0.9027** | **33.29/0.9382** | **39.54/0.9792** |
| | 36 | 28 | 38.25/0.9617 | 33.98/0.9210 | 32.34/0.9019 | 33.16/0.9373 | 39.48/0.9785 |

Table 4: The validation of DW-SwiFFN effectiveness, where MR indicates the MLP ratio, CF, SG, and DSF represents ConvFFN, SwiGLU, and DW-SwiFFN, respectively.

| CF | SG | DSF | MR | Set5 | Set14 | B100 | Urban100 | Manga109 |
|---|---|---|---|---|---|---|---|---|
| ✓ | - | - | 2.0 | 38.18/0.9616 | 33.89/0.9206 | 32.32/0.9013 | 32.97/0.9356 | 39.28/0.9785 |
| ✓ | - | - | 1.5 | 38.16/0.9614 | 33.86/0.9203 | 32.32/0.9011 | 32.96/0.9358 | 39.27/0.9785 |
| - | ✓ | - | 1.5 | 38.22/0.9617 | 33.92/0.9207 | 32.34/0.9017 | 32.99/0.9361 | 39.30/0.9786 |
| - | - | ✓ | 1.5 | **38.23/0.9619** | **34.07/0.9213** | **32.37/0.9024** | **33.08/0.9369** | **39.38/0.9792** |

**DW-SwiFFN.** We ablate feed-forward designs on DPMFormer-B by replacing its FFN with ConvFFN (ratios 2.0 and 1.5), SwiGLU, and the proposed DW-SwiFFN; unless noted, the MLP ratio is 1.5 to keep the model lightweight. As shown in Table 4, reducing the ConvFFN ratio from 2.0 to 1.5 has negligible impact, while DW-SwiFFN attains the best reconstruction quality.

**RMSNorm** *vs.* **LayerNorm.** We replace LayerNorm with RMSNorm throughout DPMFormer-B (with DW-SwiFFN). As shown in Table 5, RMSNorm yields small, dataset-dependent gains—+0.11 dB on Urban100 and +0.15 dB on Manga109—together with lower normalization overhead, motivating its use in our lightweight setting; similar trends are observed on HiT-SR.

Table 5: Performance differences of DPMFormer-DSF under different normalization schemes.

| Method | #Para. | Norm | Set5 | Set14 | B100 | Urban100 | Manga109 |
|---|---|---|---|---|---|---|---|
| HiT-SR | 847K | LayerNorm | 38.21/0.9612 | 33.83/0.9206 | 32.33/0.9021 | 33.07/0.9369 | 39.34/0.9787 |
| | 843K | RMSNorm | 38.27/0.9616 | 34.05/0.9211 | 32.37/0.9028 | 33.14/0.9376 | 39.40/0.9791 |
| DPMFormer-DSF | 829K | LayerNorm | 38.23/0.9615 | 34.07/0.9213 | 32.37/0.9024 | 33.08/0.9369 | 39.38/0.9792 |
| | 826K | RMSNorm | **38.29/0.9618** | **34.12/0.9219** | **32.40/0.9027** | **33.29/0.9382** | **39.54/0.9792** |

**LBi-Mamba.** To assess LBi-Mamba's impact on computation and precision, we perform an ablation in Table 6. Relative to a direct bidirectional Mamba, LBi-Mamba uses 7K fewer parameters and achieves higher performance, confirming its effectiveness within our framework.

**IEB.** We ablate IEB against two alternatives—linear fusion and no inter-branch fusion. Although these variants are more lightweight, IEB delivers superior accuracy with only marginal latency differences, validating the design as shown in Table 7.

## 5 CONCLUSION

In this work, we propose DPMFormer, an efficient dual-branch super-resolution framework that fuses state-of-the-art Transformer-based SR methods with Mamba to achieve superior reconstruction performance. Our design introduces an ETB enhanced with MCB, which uses LBi-ML and CAL to enrich and effectively integrate image features, and we additionally adopt RMSNorm and a depthwise-enhanced SwiFFN to simplify normalization and feed-forward computation. Ablations indicate modest improvements under our training setup. Extensive evaluations demonstrate the effectiveness of DPMFormer, under matched output size $720 \times 1280$ and reported latency on RTX 4090, DPMFormer delivers leading or competitive quality with lower MACs and params than strong baselines.

Table 6: The performance differs of DPMFormer using LBi-Mamba and Bi-Mamba.

| Method | #Para. | MACs | Set5 | Set14 | B100 | Urban100 | Manga109 |
|---|---|---|---|---|---|---|---|
| DPMFormer with Bi-Mamba | 833K | 183.4G | 38.27/0.9615 | 34.09/0.9218 | 32.39/0.9024 | 33.23/0.9378 | 39.49/0.9788 |
| DPMFormer with LBi-Mamba | 826K | 179.2G | **38.29/0.9618** | **34.12/0.9219** | **32.40/0.9027** | **33.29/0.9382** | **39.54/0.9792** |

Table 7: Impact of inter-branch fusion on PSNR/SSIM and Latency.

| Fusion method | #Para. | MACs | Running time(ms) | B100 | Urban100 | Manga109 |
|---|---|---|---|---|---|---|
| None | 807K | 175.3G | 636 | 32.28/0.9017 | 33.18/0.9366 | 39.33/0.9782 |
| Linear | 815K | 177.6G | 652 | 32.36/0.9024 | 33.21/0.9373 | 39.40/0.9787 |
| IEB | 826K | 179.2G | 668 | **32.40/0.9027** | **33.29/0.9382** | **39.54/0.9792** |

## ETHICS STATEMENT

This work develops a lightweight image super-resolution model and *does not involve new data collection, human subjects, or personally identifiable information*. We train on standard public datasets and evaluate on five widely used SR benchmarks; all assets are used under their respective licenses. The method does not target biometric identification or demographic inference and introduces no sensitive attributes in either training or evaluation. We do not position the model for forensic, medical, or safety-critical use without domain-specific validation. We are not aware of any material ethical concerns specific to this work.

## REPRODUCIBILITY STATEMENT

We will release our full codebase upon acceptance, including training/evaluation scripts, configuration files, model definitions, data preparation utilities, and scripts for parameter and MACs accounting and latency measurement. Checkpoints for all reported scales will also be provided to reproduce the main tables and figures end-to-end.

The main paper specifies all details required for reproduction: datasets and evaluation protocol (PSNR/SSIM on the luminance channel with standard crop), degradation settings, patch sizes, window sizes, the number of ETBs per stage, per-stage channel widths and the ACLinear schedule ($C_1 \leftrightarrow C_2$), CAL window size, DW-SwiFFN ratio, and LBi-Mamba settings (channel split and sequence reversal). We further disclose optimizer, learning-rate schedule, batch size, total training steps/epochs, and data augmentation.

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

# A APPENDIX

## A.1 OVERALL ARCHITECTURE

As with other lightweight super-resolution (SR) models, our proposed architecture comprises three fundamental components: a shallow feature extraction module utilizing $3 \times 3$ convolutional layers, a dual-branch deep feature extraction module, and an upsampling reconstruction module employing Pixel Shuffle operations. Given an input image $\mathcal{G}_{LR}$, the output from the shallow feature extraction module is formulated as:

$$\mathbf{F}_0 = \mathrm{H}_{sf}(\mathcal{G}_{LR}) \tag{8}$$

where $\mathrm{H}_{sf}(\cdot)$ denotes the shallow feature extraction function implemented through $3 \times 3$ convolutional operations, and $\mathbf{F}_0$ represents the extracted shallow features subsequently processed by the dual-branch deep feature extraction module.

Then, we divide $\mathbf{F}_0$ into two equal parts along the channel dimension: one is forwarded through the Transformer path, while the other is processed by the Mamba path. This operation can be formulated as:

$$\mathbf{F}_{t_0}, \ \mathbf{F}_{m_0} = \mathrm{S}(\mathbf{F}_0) \tag{9}$$

where $\mathrm{S}(\cdot)$ denotes the channel-wise split function that partitions the input feature into two sub-features of equal dimensionality.

The dual-branch deep feature extraction module consists of two distinct yet complementary branches: a Transformer Branch comprising multiple Enhanced Transformer Blocks (ETBs) and a Mamba Branch constructed from several Mamba Cross Blocks (MCBs). The Transformer Branch utilizes Window-based Attention mechanisms designed explicitly to capture fine-grained local spatial features effectively. Conversely, the Mamba Branch integrates Mamba modules optimized for

efficiently modeling long-range global dependencies. Additionally, the MCB incorporates a Cross-Attention mechanism strategically developed to adaptively fuse complementary feature representations derived from both Transformer and Mamba branches, thereby significantly enhancing the model's representational capability and overall performance.

To facilitate efficient cross-branch interaction, we introduce the Inter-branch Exchange Bridges (IEB), which is symmetrically applied in both Transformer-to-Mamba and Mamba-to-Transformer directions. Each IEB consists of a patch unembedding layer, a convolutional enhancement module (ConvBlockD), and a patch embedding layer. The ConvBlockD module comprises two stacked $1 \times 1$ convolutional layers with an intermediate GELU activation, enabling effective refinement of spatial and semantic features. For the Transformer branch, a Dual Feature Extraction (DFE) module is employed within the IEB to emphasize local details and structural cues before integration. For the Mamba branch, a single linear projection is used to align the global representations with the Transformer feature space. This design preserves computational efficiency while enhancing bidirectional information flow and ensuring better feature alignment for accurate reconstruction.

Specifically, let the number of stacked blocks be H, and let $\mathbf{F}_t$ and $\mathbf{F}_m$ denote the intermediate feature representations from the Transformer and Mamba branches, respectively. At each layer $i \in \{1, 2, \ldots, 4\}$, the forward computation proceeds as follows:

The input to the ETB is updated using shallow features or the fused result from the previous layer:

$$\mathbf{F}_{t_i} = \mathrm{H}_{etb}^i(\mathbf{F}_{t_{i-1}} + \delta \cdot \mathrm{H}_{ieb}^{i-1}(\mathbf{F}_{m_{i-1}})) \tag{10}$$

where $\mathrm{H}_{etb}^i(\cdot)$ denotes the $i$-th ETB, $\mathrm{H}_{ieb}^{i-1}(\cdot)$ denotes the $i-1$-th IEB which exchanges information from MCB to ETB, and $\delta$ is an indicator function that activates the residual connection only when $i > 1$.

The Mamba Cross Block processes the current Mamba features with the transformed ETB output:

$$\mathbf{F}_{m_i} = \mathrm{H}_{mcb}^i(\mathbf{F}_{m_{i-1}}), \ \mathrm{H}_{ieb}^i(\mathbf{F}_{t_i}) \tag{11}$$

where $\mathrm{H}_{mcb}^i$ denotes the $i$-th MCB which integrates Mamba and cross-attention mechanisms, $\mathrm{H}_{ieb}^{i-1}(\cdot)$ denotes the $i-1$-th IEB which exchanges information from ETB to MCB.

After the final stage of deep feature extraction, we obtain two refined feature streams: $\mathbf{F}_t$ from the Transformer branch and $\mathbf{F}_m$ from the Mamba branch. Before reconstruction, residual connections are applied to both:

$$\mathbf{F}_t' = \mathrm{H}_{tc}(\mathbf{F}(t)) + \mathbf{F}_{t_0} \tag{12}$$

$$\mathbf{F}_m' = \mathrm{H}_{mc}(\mathbf{F}_m) + \mathbf{F}_{m_0} \tag{13}$$

where $\mathrm{H}_{tc}$ and $\mathrm{H}_{mc}$ denotes a $3 \times 3$ convolutional layer used to refine the Transformer features and Mamba features. The terms $\mathbf{F}(t_0)$ and $\mathbf{F}(m_0)$ are shallow features from each branch used for residual learning.

The two enhanced features are concatenated along the channel dimension and fed into the upsampling module:

$$\mathcal{G}_{SR} = \mathrm{U}([(\mathbf{F}_t', \ \mathbf{F}_m')]), \tag{14}$$

where, $[\cdot, \cdot]$ denotes the concatenation operation along the channel dimension and $\mathcal{U}(\cdot)$ denotes the Upsampler, which consists of a $3 \times 3$ convolution that increases the number of channels to $(r^2 \cdot C)$, followed by a PixelShuffle operation with upscale factor $r$ to reconstruct the final high-resolution image $\mathcal{G}_{SR}$.

This design ensures that both local and global contextual features contribute directly to the final super-resolved output.

## A.2 ENHANCED TRANSFORMER BLOCK

The Enhanced Transformer Block (ETB) consists of six stacked Enhanced Transformer Layers (ETLs) and a residual connection. To enhance the receptive field, each ETL employs window-based attention with variable window sizes, we use a base $8 \times 8$ window and set $\alpha_i \in \{0.5, 1, 2, 4, 6, 8\}$ for the $i$-th ETL. The window size for each block is dynamically adjusted as follows:

$$w_H = \alpha_i \cdot h, \quad w_W = \alpha_i \cdot w \tag{15}$$

Figure 5: Comparisons between different SOTA methods, where local attribution maps (LAM) and informative areas (Info. Area) are displayed.

where $h$ and $w$ denote the initial window height and width, and $\alpha_i$ is the scaling factor at the $i$-th block.

Let the input feature be denoted as $\mathbf{F}_{t_{i-1}}$, the forward propagation within the ETB can be formulated as:

$$\mathbf{F}_{t_i} = \mathrm{L}_c(\mathrm{L}_{etl}^{\times 6}(\mathbf{F}_{t_{i-1}}), \ (w_H, w_W)) + \mathcal{F}(t_{i-1}) \tag{16}$$

where $\mathrm{L}_{etl}^{\times 6}$ represents six consecutive ETLs and $\mathrm{L}_c$ is a $3 \times 3$ convolutional layer applied after the ETLs.

## A.3 Enhanced Transformer Layer

Each Enhanced Transformer Layer (ETL) is composed of three core components: Root Mean Square Layer Normalization (RMSNorm), Spatial Channel Correlation (SCC), and a Depthwise Convolution-enhanced SwiFFN module (DW-SwiFFN). Given an input feature $\mathcal{F}_{in} \in \mathbb{R}^{B,H,W,C}$, we first apply a window-based reshape operation:

$$\mathbf{F}'_{in} = O_{reshape}(\mathbf{F}_{in}), \quad \mathbf{F}'_{in} \in \mathbb{R}^{B',w_H w_W,C} \tag{17}$$

where $O_{reshape}(\cdot)$ denotes the spatial window partitioning function.

Next, the reshaped feature is processed by the SCC module:

$$\mathbf{F}_{scc} = M_{scc}(\mathbf{F}'_{in}) \tag{18}$$

where $M_{scc}(\cdot)$ refers to the Spatial Channel Correlation operation.

We then reverse the reshape and apply RMSNorm followed by a residual connection:

$$\mathbf{F}_{att} = M_{rms}(O_{reshape}(\mathbf{F}_{scc})) + \mathbf{F}_{in} \tag{19}$$

where $M_{rms}(\cdot)$ denotes the RMSNorm operation.

Finally, the DW-SwiFFN module and another RMSNorm are applied, with an additional residual connection:

$$\mathbf{F}_{out} = M_{rms}(M_{ffn}(\mathbf{F}_{att})) + \mathbf{F}_{att} \tag{20}$$

where $M_{ffn}(\cdot)$ denotes the depthwise convolution-augmented feed-forward network (DW-SwiFFN).

## A.4 Mamba Cross Block

The proposed Mamba Cross Block (MCB) is composed of a sequence of three Lightweight Bidirectional-Mamba Layers (LBi-MLs), each followed by a Cross-Attention Layer (CAL), together with residual connections to enable stable feature flow.

Let $\mathcal{F}(m_{i-1})$ denote the input from the previous MCB and $\mathcal{F}(t_{i-1})$ denote the features from the corresponding ETB after Fusion Adaptive Block (FAB) transformation. The MCB output is computed as:

$$\mathbf{F}(m_i) = L_{cmcl}(\mathbf{F}(m_{i-1}), \mathbf{F}(t_{i-1})) + \mathbf{F}(m_{i-1}) \tag{21}$$

where $L_{cmcl}(\cdot, \cdot)$ represents the composite operation of LBi-MLs and the Cross-Attention Layer (CAL), enabling effective bidirectional modeling and inter-branch feature fusion.

## A.5 Lightweight Bidirectional-Mamba Layer

The Lightweight Bidirectional-Mamba Layer (LBi-ML) is designed to efficiently model long-range dependencies via selective state-space dynamics while preserving bidirectional context. It consists of three main components: Lightweight Bidirectional-Mamba (LBi-Mamba), RMSNorm and DW-SwiFFN.

The core component of this module is the LBi-Mamba, whose internal computation process is detailed in Algorithm 1. The remaining forward logic, including padding, residual connections, and feed-forward operations, follows the structure illustrated in Figure 1 (b). Given an input feature $\mathbf{F}_{in}$, and auxiliary feature $\mathbf{F}_{af}$ from ETB, the process of LBi-ML can be expressed as:

$$\mathbf{F}_{lbm} = M_{rms}(M_{lbm}(\mathbf{F}_{in} + \mathbf{F}_{af})) + \mathbf{F}_{in} \tag{22}$$

where $M_{lbm}(\cdot)$ refers to the LBi-Mamba.

$$\mathbf{F}_{out} = M_{rms}(M_{ffn}(\mathbf{F}_{lbm})) + \mathbf{F}_{lbm}, \tag{23}$$

where $\mathbf{F}_{out}$ is the output of the LBi-ML.

---

**Algorithm 1** Forward of LBi-Mamba

---

**Require:** $\mathbf{F}_{in} \in \mathbb{R}^{B \times C \times H \times W}$, auxiliary $\mathbf{F}_{af}$, group number $G$, sequence length $L = H \cdot W$
**Ensure:** $\mathbf{F}_{out}$

1: $\tilde{\mathbf{F}} \leftarrow \mathbf{F}_{in} + \mathbf{F}_{af}$
2: $\mathbf{F} \leftarrow \mathrm{P}_{in}(\mathbf{F})$ {linear proj. for scan/skip}
3: $(\mathbf{x}, \mathbf{z}) \leftarrow \mathrm{Split}(\mathbf{F})$
4: **Segment:** reshape $\mathbf{x}$ into $\{\mathbf{x}^{(g)}\}_{g=1}^G$
5: **Build single bidirectional sequence:** $\hat{\mathbf{x}} \leftarrow [\,\mathbf{x}, \mathrm{rev}(\mathbf{x})\,]$
6: $\hat{\mathbf{x}} \leftarrow \mathrm{Conv1D}(\hat{\mathbf{x}}); \quad (\Delta t, \mathbf{B}, \mathbf{C}) \leftarrow \mathrm{P}_{scan}(\hat{\mathbf{x}})$
7: $y \leftarrow \mathrm{SelectiveScan}(\hat{\mathbf{x}}, \Delta t, \mathbf{A}, \mathbf{B}, \mathbf{C}, \mathbf{D})$ {one pass over fwd+rev}
8: **Recover order:** split $\mathbf{y} = [\mathbf{y}_\rightarrow, \mathbf{y}_\leftarrow]; \mathbf{y}_\leftarrow \leftarrow \mathrm{rev}(\mathbf{y}_\rightarrow)$; interleave/concat to match segments
9: $\mathbf{F}_{out} \leftarrow \mathrm{P}_{out}([\,\mathbf{y}, \mathbf{z}\,])$
10: **return** $\mathbf{F}_{out}$

---

*Complexity note:* selective-scan over $\hat{\mathbf{x}}$ is $O(L \cdot C)$; $\mathrm{P}_{in}/\mathrm{P}_{out}$ are $O(L \cdot C)$; grouping and flips are linear in $L$ with small constants.

---

**Algorithm 2** Forward Pass of Cross Attention

---

**Input**: Input feature $\mathbf{F}_{in}$, Auxiliary feature $\mathbf{F}_{af}$
**Output**: Output feature $\mathbf{F}_{out}$

1: Extract key and value: $[\mathbf{k}, \mathbf{v}] \leftarrow \mathrm{DFE}(\mathbf{F}_{in})$
2: Extract query: $\mathbf{q} \leftarrow \mathrm{DFE}(\mathbf{F}_{af})$
3: Partition $\mathbf{q}, \mathbf{k}, \mathbf{v}$ into windows of size $(w_H, w_W)$
4: **for all** each window **do**
5:     Normalize: $\mathbf{q} \leftarrow \mathrm{Norm}(\mathbf{q}), \quad \mathbf{k} \leftarrow \mathrm{Norm}(\mathbf{k})$
6:     Compute attention logits with relative bias:

$$\mathbf{att} = \frac{\mathbf{q} \cdot \mathbf{k}^T}{scale}$$

7:     Compute attention weights: $\mathbf{att} \leftarrow \mathrm{softmax}(\mathbf{att})$
8:     Attend to values: $o \leftarrow \mathbf{att} \cdot \mathbf{v}$
9: **end for**
10: Merge windows and apply projection: $\mathbf{F}_{out} \leftarrow \mathtt{Proj}(o)$
11: **return** $\mathbf{F}_{out}$

---

A.6   CROSS ATTENTION LAYER

The Cross-Attention Layer (CAL) is designed to adaptively fuse query features from transformer path and key-value features from mamba path under a windowed attention scheme. CAL consists of three main components: Cross Attention (CrA), RMSNorm, and DW-SwiFFN. The above ratio policy applies to ETL. By contrast, CAL uses a fixed $8 \times 8$ window.

The core component of this module is the CrA operation, whose attention process is detailed in Algorithm 2. The rest of the forward computation, including image padding, residual connections, and the feed-forward module, follows the layout shown in Figure 1 (c).

Given an input feature $\mathbf{F}_{in}$ and an auxiliary cross-path feature $\mathbf{F}_{af}$, the process of CrA-based attention-enhanced Block can be expressed as:

$$\mathbf{F}_{cal} = \mathrm{M}_{rms}(\mathrm{M}_{cra}(\mathbf{F}_{in}, \mathbf{F}_{af})) + \mathbf{F}_{in}, \tag{24}$$

where $\mathrm{M}_{cra}(\cdot)$ refers to the CrA module, and $\mathrm{M}_{rms}(\cdot)$ denotes the RMS normalization layer.

$$\mathbf{F}_{out} = \mathrm{M}_{rms}(\mathrm{M}_{ffn}(\mathbf{F}_{cra})) + \mathbf{F}_{cra}, \tag{25}$$

where $\mathbf{F}_{out}$ is the final output feature from the CAL Block.

## A.7 Loss Function

Similar to most lightweight super-resolution models, we employ the $\ell_1$ loss to supervise network training. Let $I_{SR}$ denote the super-resolved image predicted by the network and $I_{HR}$ the corresponding highFLP-resolution ground truth. The loss function is defined as:

$$L_1 = \|I_{SR} - I_{HR}\|_1 \tag{26}$$

This formulation encourages pixel-wise consistency and facilitates stable convergence during training.

## A.8 Information Aggregation Capability Analysis

As illustrated in the Figure 5, we compare the Local Attribution Maps (LAM) and Informative Areas (Info. Area) of several models, including SRFormer, MambaIR, HiT-SR, CATANet, and DPMFormer .

Analysis of the First Region, it is evident that the LAM of MambaIR is primarily concentrated around the target region, indicating a strong focus on local information aggregation. In contrast, HiT-SR and DPMFormer exhibit broader LAM activations, suggesting an enhanced ability to capture long-range dependencies and contextual features. Notably, the LAM response of DPMFormer is not only extensive but also demonstrates higher intensity and more precise coverage of both the target and its relevant contextual regions compared to HiT-SR. In the visualization of informative areas, HiT-SR and DPMFormer both present larger, more uniformly distributed high-response areas, indicating better identification and aggregation of target-related pixels. However, DPMFormer stands out by exhibiting the widest and most coherent informative area among all models, highlighting its superior capacity for comprehensive information integration. In comparison, the informative areas of MambaIR and SRFormer remain relatively limited and concentrated.

Analysis of the Second, similar patterns can be observed in other target regions. DPMFormer consistently delivers the most extensive and continuous informative areas, efficiently encompassing the target and its contextual surroundings. This enables DPMFormer to achieve a more holistic perception of complex textures and structural details, surpassing the performance of HiT-SR, which, is less effective in integrating global contextual information to the same extent. CATANet and SRFormer achieve moderate coverage, balancing local and global information, but their ability to aggregate information in complex scenarios is evidently inferior to DPMFormer.

In summary, DPMFormer demonstrates the most advantageous information aggregation capability across all examined regions. Its extensive, high-intensity LAM activations and the largest, most coherent informative areas clearly surpass those of other models—including HiT-SR—allowing DPMFormer to fully exploit long-range dependencies and intricate structural cues. This ultimately leads to superior performance in super-resolution reconstruction, especially in scenarios that demand rich contextual integration and accurate detail recovery. By contrast, while HiT-SR also shows strong aggregation abilities, it does not reach the same level of coverage and precision as DPMFormer. MambaIR, CATANet, and SRFormer, which rely more on local features.

## A.9 Comparative Analysis of Texture Detail Restoration

According to the output feature visualizations of each branch in Figure 6, a clear distinction can be observed between the ETB and the MCB in their capability to recover texture details.

**Texture Representation of ETB.** The ETB is specifically designed to enhance the representation of fine-grained textures and local structural details. As seen from the first to the fourth ETB output features, this branch consistently preserves high-frequency information and subtle texture variations.

The features generated by ETB retain sharp edges and intricate patterns, closely approximating the structural details of the original image. This is particularly evident in regions with rich textures or complex patterns, where ETB maintains clarity and continuity, effectively avoiding over-smoothing.

**Texture Representation of MCB.** In contrast, the MCB focuses on capturing broader contextual information and semantic structures. The output features of the MCB exhibit a more homogeneous and smoothed appearance, which helps in modeling global spatial relationships but often results in a loss of fine texture details.

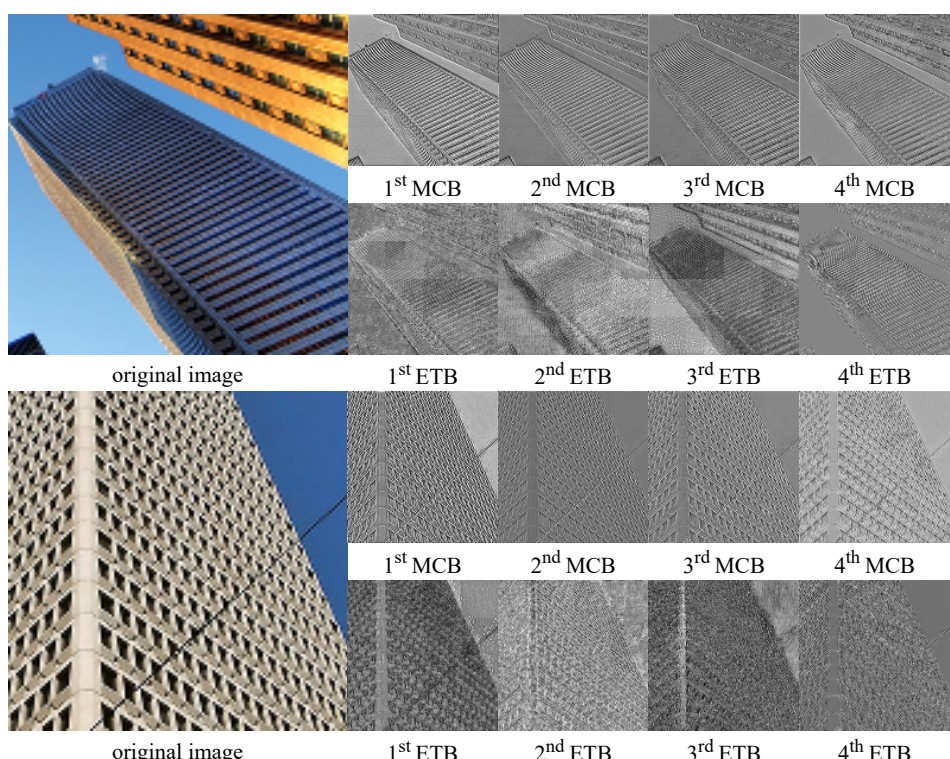

Figure 6: Comparison of texture detail restoration capabilities between ETB and MCB.

While the MCB provides robust contextual understanding and stabilizes the overall image structure, it tends to suppress local texture fluctuations. Consequently, MCB may fail to recover subtle details in highly textured regions, leading to a relatively smoother reconstruction.

In summary, ETB demonstrates a clear advantage in restoring fine texture details and preserving local structures, making it particularly effective for super-resolution tasks requiring high-frequency detail recovery. MCB, on the other hand, excels at maintaining global consistency and suppressing noise, but may not be as effective as ETB in recovering intricate textures. The combination of ETB and MCB leverages the strengths of both branches: ETB contributes sharpness and texture fidelity, while MCB ensures semantic coherence and stability across the image.

## A.10 ENHANCEMENTS ACHIEVED BY DPMFORMER

Figure 7 illustrates how DPMFormer enriches a variety of low-resolution (LR) inputs. For each split image, the left side reproduces the LR content (up-scaled only for display), while the right side shows the result obtained with DPMFormer.

**Linear-Structure Enhancement.** Straight edges such as building contours, window mullions, and bridge columns are precisely reconstructed. This is evident in "img_022", where roof finials and masonry joints regain geometric sharpness, and in "0814", where the observation deck's curvature and vertical supports are rendered without aliasing.

**Text and Line-Art Enhancement.** Thin strokes, lettering, and comic outlines become crisp and continuous. In "comic", facial lines and patterned garments recover clear tonal boundaries, while in

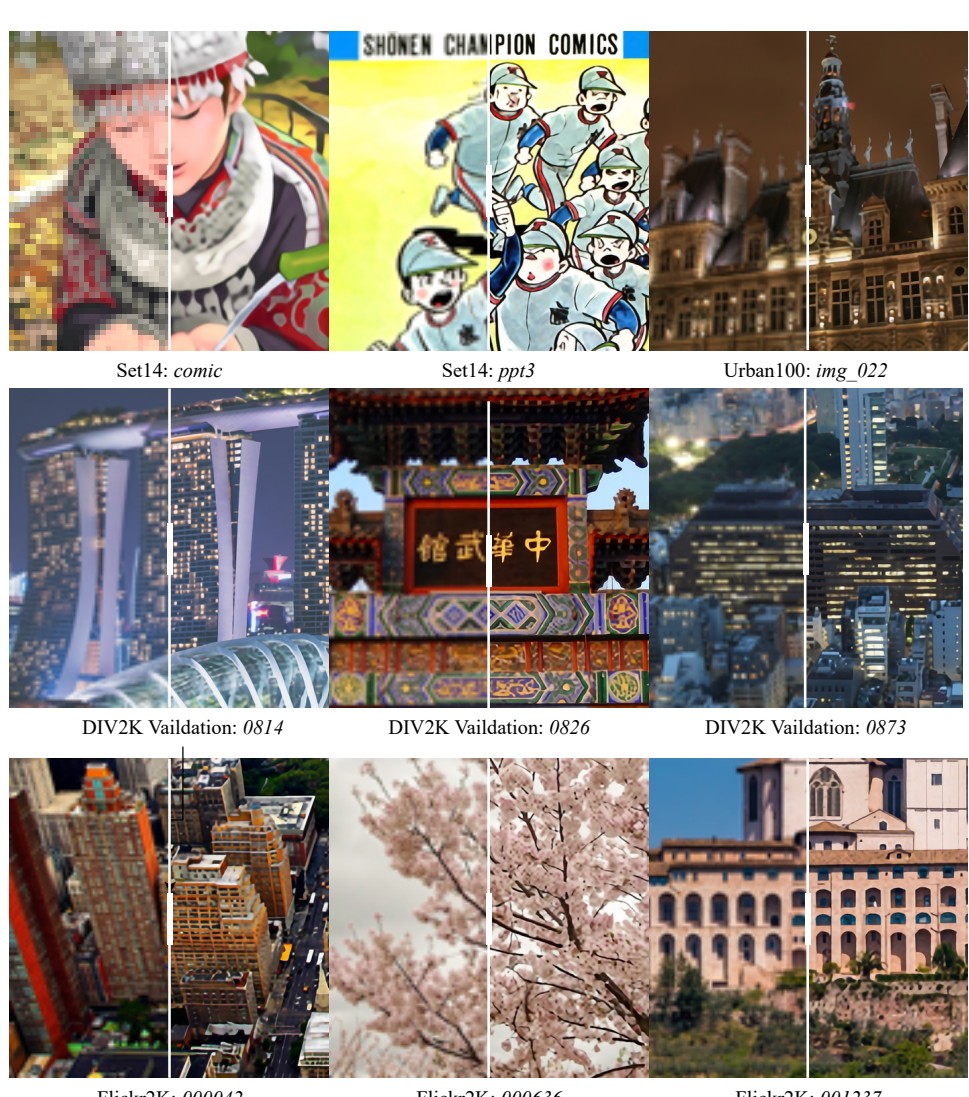

Figure 7: Visual comparison between bicubic-interpolated LR inputs (left in each pair, for reference only) and the proposed DPMFormer outputs (right). DPMFormer delivers markedly sharper edges, richer textures, and fewer artifacts across cartoon, architectural, night-scene, and natural imagery.

"ShimatteIkouze_vol26" character outlines, screen-print textures, and facial expressions are sharply reinstated.

**Fine Texture and Pattern Enhancement.** High-frequency details, bricks, floral petals, ornamental carvings are faithfully restored. "0826" shows brush-stroke fidelity and multicolor carvings on a Chinese gate; "000636" reveals individual cherry-blossom petals with smooth gradients; "000042" and "001237" exhibit well-defined brick patterns, rooftop edges, and stone textures, all while preserving global symmetry.

Across these diverse scenes, cartoons, cultural landmarks, night skylines, and natural imagery, DPM-Former consistently restores high-frequency details, sharp contours, and faithful textures that are either lost or severely blurred in the original LR frames.

## A.11 MORE COMPREHENSIVE EFFICIENT ANALYSIS

We further provide a more comprehensive and in-depth comparison across models efficient. In the main paper, we already report the number of parameters, MACs, and inference time. Here, we additionally compare the throughput and average allocated GPU memory of different models. Specifically, we benchmark SwinIR-L, SRFormer, HiT-SRF, CATANet, and MambaIRv2, and summarize the results in Table 8. We also include several larger image restoration (IR) models for reference (Zamir et al., 2022; Shi et al., 2025). Note that Restormer does not provide a super-resolution (SR) variant, for a fair comparison, we remove its up- and down-sampling modules so that its backbone is aligned with the SR setting.

Table 8: Comparison of throughput and memory allocation (Mem. Alloc.) with SOTA models when super-resolving 1280×720 images at a 2× upscaling on a RTX 4090. ↑ and ↓ indicate that higher and lower values are better, respectively.

| Method | Params (K) | MACs (G) | FLOPs (G) | Throughput (img/s) ↑ | Mem. Alloc. (MB) ↓ |
|---|---|---|---|---|---|
| Restormer | 26,127 | 544.5 | 1089.0 | 0.41 | 107.10 |
| VMambaIR | 10,498 | 881.6 | 1763.1 | 1.19 | 93.75 |
| SwinIR | 910 | 244.4 | 488.9 | 0.31 | 74.46 |
| SRFormer | 853 | 236.2 | 472.4 | 0.27 | 72.34 |
| HiT-SRF | 847 | 226.5 | 452.1 | **2.35** | **58.58** |
| CATANet | 477 | 164.6 | 329.3 | 0.70 | 55.13 |
| MambaIRv2 | 774 | 286.3 | 572.6 | 0.89 | 58.52 |
| **DPMFormer** | 826 | 179.2 | 358.4 | *2.24* | *57.12* |

Table 8 reports the model complexity and runtime characteristics of our DPMFormer against recent SR baselines. The upper part of the table contains high–capacity models, while the lower part focuses on lightweight designs with fewer than 1M parameters. DPMFormer uses only 0.83M parameters, which is $31.6\times$ smaller than Restormer and $12.7\times$ smaller than VMambaIR. In terms of computation, our model requires 179.2G MACs, reducing FLOPs by $3.0\times$ and $4.9\times$ relative to Restormer and VMambaIR, respectively. Despite the much smaller budget, DPMFormer achieves a throughput of 2.24 img/s, which is $5.5\times$ higher than Restormer and $1.9\times$ higher than VMambaIR, while also cutting the peak memory allocation by roughly 50 MB and 37 MB, respectively.

Within the lightweight regime, DPMFormer attains a very favorable trade-off between computation and speed. Compared to SwinIR and SRFormer, our model uses a similar number of parameters but reduces FLOPs by about $25\%$–$30\%$, and yet improves throughput by $7.2\times$ and $8.3\times$, respectively, while also requiring 15-17 MB less memory. CATANet has slightly lower MACs, but its throughput is only 0.70 img/s, DPMFormer is $3.2\times$ faster at a comparable memory footprint. Relative to the stronger Mamba-based baseline MambaIRv2, DPMFormer decreases FLOPs by $1.6\times$, achieves $2.5\times$ higher throughput, and uses a similar amount of memory. Finally, compared with HiT-SRF, which is the fastest baseline, DPMFormer attains comparable throughput while cutting FLOPs by about $20\%$ and keeping memory usage almost unchanged.

Overall, DPMFormer delivers competitive or superior runtime efficiency under a strict parameter and FLOP budget, it achieves heavy-model throughput with two orders of magnitude fewer parameters, and consistently outperforms prior lightweight SR networks in the joint space of MACs, throughput and memory allocation.

## A.12 ADDITIONAL DEGRADATION

### A.12.1 SINGLE IMAGE DEHAZING

**Experimental setup.** To further assess the generalization ability of DPMFormer beyond bicubic super-resolution, we adapt our model to single-image dehazing on the RESIDE $\beta$ benchmark (Li et al., 2019). Unless otherwise stated, we follow the same optimization settings as in the

Table 9: Comparison of DPMFormer with representative single-image dehazing methods on the SOTS (Outdoor) subset.

| Method | Source | SOTS (Outdoor) | | | |
| | | Perceptual | | Distortion | |
| | | FID↓ | LPIPS↓ | PSNR↑ | SSIM↑ |
|---|---|---|---|---|---|
| AOD-Net (Li et al., 2017) | ICCV'17 | 29.13 | 0.0639 | 19.66 | 0.8152 |
| MSCNN-HE (Ren et al., 2020) | IJCV'20 | 28.81 | 0.0458 | 20.46 | 0.8389 |
| LD-Net (Ullah et al., 2021) | TIP'21 | 39.16 | 0.0749 | 21.94 | 0.8539 |
| SG-Net (Hong et al., 2023) | ACCV'22 | 23.01 | 0.0493 | 25.21 | 0.9234 |
| SDA-GAN (Dong et al., 2023) | TCyb'23 | 15.44 | 0.0395 | 29.77 | 0.9556 |
| RI-SCNN-Large (Zhang et al., 2025b) | TNNLS'24 | 10.69 | **0.0207** | 29.02 | 0.9490 |
| DPMFormer (Ours) | - | **10.22** | 0.0277 | **29.35** | **0.9618** |

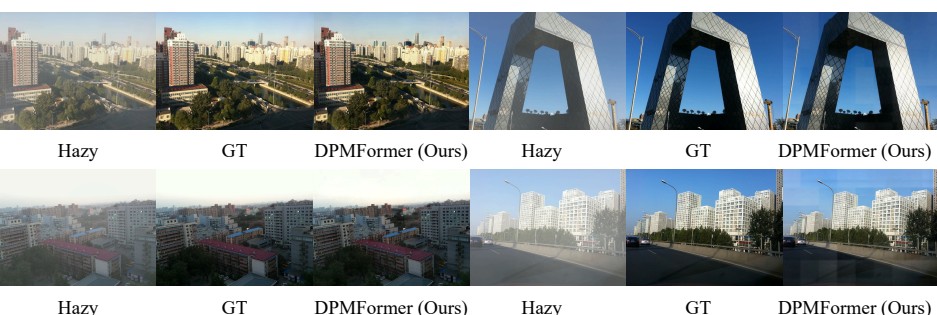

Figure 8: Qualitative dehazing results on SOTS (Outdoor). For each scene we show the hazy input, the ground-truth clear image, and the output of DPMFormer.

main super-resolution experiments, but change the learning rate to $2 \times 10^{-4}$ and replace the $\ell_1$ loss with an $\ell_2$ reconstruction loss. We directly reuse the $4\times$ SR configuration of DPMFormer without any architectural modification, and train the network on the Outdoor Training Set (OTS) of RESIDE, while evaluating on the Outdoor subset of SOTS. Training is performed for 20,000 iterations, and the learning rate is halved every 5,000 iterations.

**Baselines and metrics.** We compare our dehazing variant of DPMFormer with several representative CNN- and GAN-based methods, AOD-Net (Li et al., 2017), MSCNN-HE (Ren et al., 2020), LD-Net (Ullah et al., 2021), SG-Net (Hong et al., 2023), SDA-GAN (Dong et al., 2023), and the recent retina-inspired spiking CNN RI-SCNN-Large (Zhang et al., 2025b). Following recent dehazing literature, we report both perceptual quality metrics, FID and LPIPS and distortion-oriented measures, PSNR and SSIM.

**Results.** As summarized in Table 9, our DPMFormer achieves the best FID and PSNR on SOTS-Outdoor, and also attains the highest SSIM among all competitors, despite being originally designed for super-resolution rather than dehazing. Compared to classical supervised CNN baselines, DPM-Former substantially improves both perceptual quality and distortion metrics, especially in terms of FID and SSIM, indicating fewer artifacts and better structural consistency. Relative to recent GAN-or spiking-based approaches, our model delivers competitive or superior perceptual scores while slightly surpassing them in PSNR/SSIM, showing that the proposed dual-path Mamba–Transformer design transfers well to haze removal without task-specific architectural tuning.

Table 10: Comparison of DPMFormer with representative LLIE methods on LOL-v2-real. Best results are highlighted in **bold** and the second best are *italic*.

| Method | Source | LOL-v2-real | | | |
|---|---|---|---|---|---|
| | | RMSE↓ | LPIPS↓ | PSNR↑ | SSIM↑ |
| Retinex-Net (Wei et al., 2018) | BMVC'18 | 20.21 | 0.436 | 16.41 | 0.640 |
| MIRNet (Zamir et al., 2020) | ECCV'20 | 12.03 | 0.250 | 20.36 | 0.782 |
| Retinexformer (Cai et al., 2023) | ICCV'23 | **8.35** | **0.131** | 21.23 | *0.838* |
| MambaIR (Guo et al., 2025b) | ECCV'24 | *9.33* | 0.258 | *21.25* | 0.831 |
| DPMFormer (Ours) | – | 9.34 | *0.221* | **21.28** | **0.860** |

**Qualitative analysis.** Fig. 8 further illustrates the dehazing behaviour of DPMFormer on several outdoor scenes. In the first example, our method effectively removes the global veiling effect, recovers the contrast between buildings and vegetation, and produces sky colors that are close to the ground truth without introducing over-saturation. Thin structures such as window frames and bridges are clearly visible, showing that the model can recover high-frequency details instead of simply brightening the image.

In the second and third scenes, haze density varies significantly with depth. DPMFormer is able to clean distant buildings and the horizon while keeping nearby regions visually consistent, leading to depth-aware restoration, foreground objects remain natural, and the background no longer appears washed out. This indicates that the proposed dual-path design can handle long-range dependency and spatially varying degradation at the same time.

The last example highlights color fidelity. Our reconstruction removes the greenish and grayish cast from the hazy input and restores neutral road colors and realistic facade textures, closely matching the ground truth. Across all four scenes, DPMFormer produces sharp edges, clear textures and natural colors, with no obvious ringing or halo artifacts, confirming that the model generalizes well from super-resolution to haze removal in real images.

### A.12.2 REAL WORLD LOW-LIGHT IMAGE ENHANCEMENT

**Experimental setup.** We further assess the generalization ability of DPMFormer on low-light image enhancement (LLIE). Unless otherwise stated, we follow the same training protocol as in the main ×4 SR experiments, we keep the AdamW optimizer and the $\ell_1$ reconstruction loss, reset the learning rate to $2 \times 10^{-4}$, and fine-tune the ×4 SR model directly for the LLIE task. We use the LOL-v2-real dataset (Chen et al., 2021b) with the official training, testing split, train the model for 20,000 iterations, and halve the learning rate every 5,000 iterations.

**Baselines and metrics.** We compare DPMFormer with four representative supervised LLIE approaches, Retinex-Net (Wei et al., 2018), IRNet (Zamir et al., 2020), Retinexformer (Cai et al., 2023), and MambaIR (Guo et al., 2025b). As shown in Table 10, DPMFormer achieves the best PSNR and SSIM, while maintaining competitive perceptual scores, RMSE and LPIPS. In particular, our model slightly improves PSNR over both Retinexformer and MambaIR and yields the highest SSIM, indicating that the proposed dual-path design transfers well from SR to illumination correction without any task-specific architectural change.

**Qualitative analysis.** Qualitative examples in Fig. 9 further support the numerical gains. On the indoor archery scene, DPMFormer not only recovers the overall illumination, but also reconstructs sharp edges on the posters and equipment without amplifying noise in the dark background. On the outdoor corridor, our method suppresses heavy shadows and reveals structures on the floor and walls, while keeping the color of the light patches close to the ground-truth image. These results suggest that the interaction between the Mamba-based global branch and the Transformer-based local

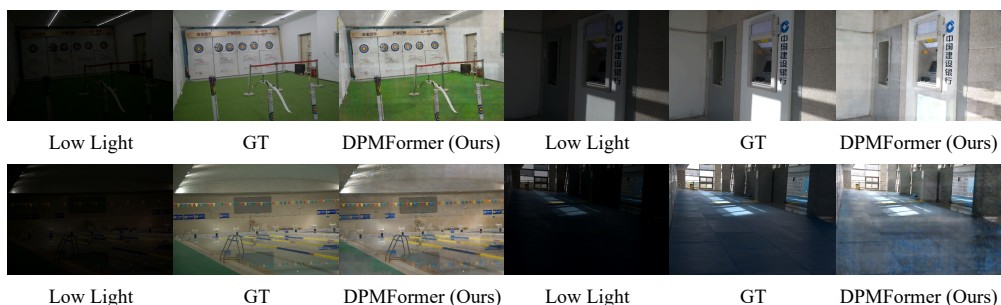

Figure 9: Qualitative LLIE results on LOL-v2-real. Our model restores global brightness while preserving local contrast and fine textures.

Table 11: Quantitative analysis of the dual-path fusion and its components on ×2 SR.

| Method | Params (K) | Set5 | Set14 | B100 | Urban100 | Manga109 |
|---|---|---|---|---|---|---|
| CPT | 942 | 38.16/0.9608 | 33.97/0.9209 | 32.33/0.9012 | 32.84/0.9341 | 39.22/0.9782 |
| CPM | 1073 | 38.15/0.9611 | 33.93/0.9205 | 32.29/0.9013 | 32.88/0.9347 | 39.26/0.9778 |
| DPMT | 831 | 38.24/0.9613 | 34.01/0.9213 | 32.36/0.9018 | 33.14/0.9374 | 39.41/0.9786 |
| DPM | 822 | 38.16/0.9610 | 33.96/0.9209 | 32.35/0.9018 | 33.04/0.9361 | 39.28/0.9784 |
| **DPMFormer** | 826 | **38.29/0.9618** | **34.12/0.9219** | **32.40/0.9027** | **33.29/0.9382** | **39.54/0.9792** |

branch is able to handle both large-scale illumination imbalance and fine-grained texture restoration in challenging low-light scenes.

### A.13 STRUCTURE AND COMPONENT ANALYSIS

#### A.13.1 QUANTITATIVE ANALYSIS OF THE DUAL-PATH DESIGN

To demonstrate the complementarity between the two paths and to isolate the effects of CAL and IEB, we construct several controlled variants. First, we build a pure Transformer baseline (CPT) by stacking four ETBs, and a pure Mamba baseline (CPM) by stacking four MCBs; both are tuned to have a similar parameter budget to DPMFormer. Second, we derive two IEB-free dual-path variants from DPMFormer: DPMT replaces the ETB→CAL feature extractor with a single linear projection, while DPM further removes the Transformer tokens from the cross-attention and replaces CAL with a vanilla windowed MHSA. All models are trained under the same setting on ×2 SR over the five standard benchmarks.

As shown in Table 11, the dual-path variants consistently outperform the single-path baselines under a comparable parameter budget. Both CPT and CPM trail behind the dual-path models on all benchmarks, confirming that combining Mamba and Transformer is beneficial rather than redundant. DPMT and DPM, which remove IEB and partly simplify CAL, already improve over CPT and CPM, but DPMFormer achieves the best PSNR/SSIM on all five datasets while using a similar number of parameters. The gains are particularly pronounced on Urban100 and Manga109, where complex structures and repeated patterns put more stress on the global–local fusion. These results indicate that the dual-path fusion is genuinely useful, and the proposed CAL, IEB design is not a trivial stacking of Mamba and Transformer blocks, but a more effective multi-way interaction mechanism.

We further examine optimization behaviour in Fig. 10, which plots the evolution of validation PSNR and gradient norms over $10^5$ iterations for DPMFormer, DPMT, and DPT. DPMFormer not only converges to the highest PSNR, but also reaches strong performance earlier and maintains a consistent

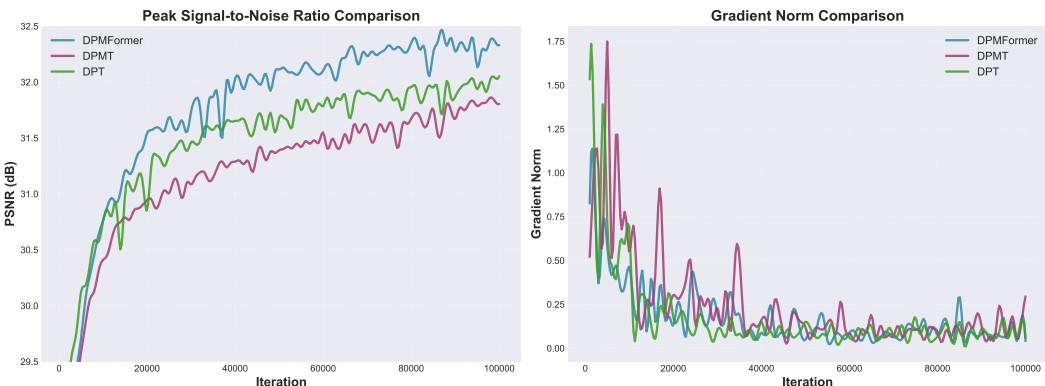

Figure 10: Training dynamics of DPMFormer and its IEB-free variants on ×2 SR. Left: validation PSNR versus training iterations. Right: $\ell_1$ gradient norm versus iterations. All models are trained under identical settings.

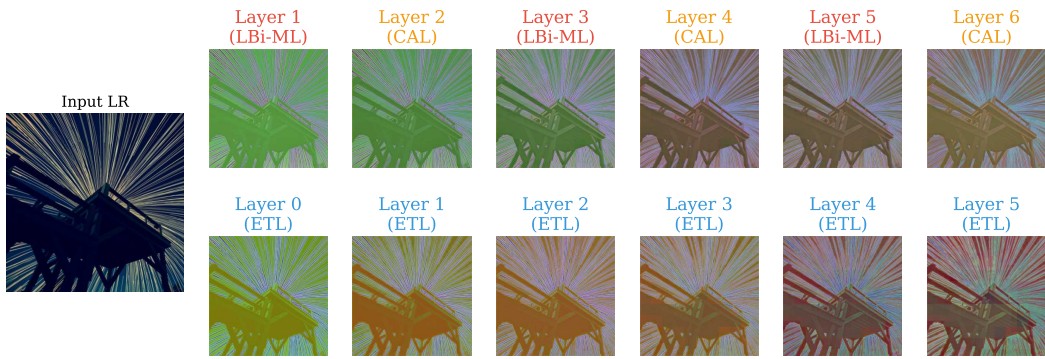

Figure 11: Comparison more details between first ETB and MCB.

margin over the two ablations throughout training. On the right, DPT and DPMT exhibit larger gradient spikes in the early iterations, whereas DPMFormer shows more moderate peaks and stabilizes faster, with smaller fluctuations in the later stage. This suggests that the full CAL and IEB coupling yields a better-conditioned optimization landscape, IEB provides a structured information flow between paths, and CAL injects Transformer features into the Mamba stream at well-chosen locations, which together regularize gradients and facilitate efficient learning. These dynamics again support our claim that DPMFormer is more than a simple Mamba and Transformer juxtaposition, and that the proposed dual-path interaction scheme is both effective and well grounded.

### A.13.2 VISUALIZATION ANALYSIS OF THE DUAL-PATH DESIGN

We conduct a more fine-grained analysis of the Layers within each ETB and MCB. For every Block, we visualize the PCA projection of each Layer and statistically summarize the information in these plots. the results for the MCB and ETB are shown in Fig. 11, Fig. 14, Fig. 17, Fig. 20.

**Shallow-layer behaviour.** Fig. 11 disentangles the roles of the three layer types in Block 1. The ETL features in the local-detail path exhibit sharp, dense responses along the radial beams and sup-

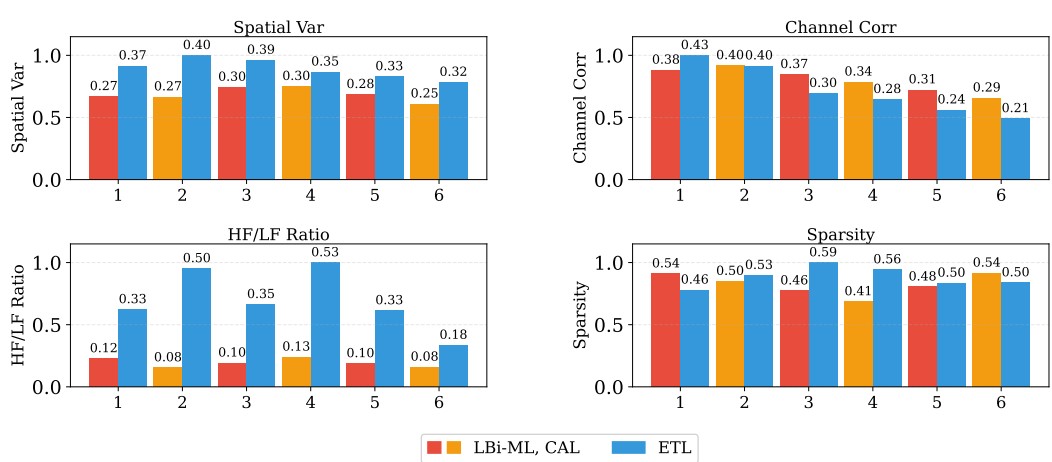

Figure 12: Normalized quantitative statistics for the first ETB and MCB.

porting structures of the tower, indicating strong sensitivity to local edges and textures from the very first block. In contrast, the LBi-ML layers in the global-context path produce much smoother activations with weak contrast, retaining only coarse information about the dominant directions and overall silhouette. The CAL layers operate on top of these LBi-ML outputs, injecting cross-branch modulation but still yielding comparatively soft responses; their feature maps accentuate intersections of rays and tower boundaries slightly more than the preceding LBi-ML layers, yet remain far less edge-enhanced than ETL.

The quantitative statistics as shown in Fig. 12 corroborate this qualitative picture. Across all six depths, the ETL layers consistently achieve higher spatial variance than both LBi-ML and CAL, reflecting stronger spatial contrast and more pronounced edge activations. When examining the HF/LF ratio, LBi-ML layers maintain a remarkably low and stable high-frequency content , showing that they encode predominantly low-frequency structure. CAL layers further suppress or only mildly boost this high-frequency energy, e.g., a slight increase at Layer 3, confirming that their main role is to reweight and redistribute information from the other branch rather than to generate new high-frequency details. Channel correlation in the global-context path stays moderately high for both LBi-ML and CAL, indicating relatively redundant channels that share similar global patterns, whereas ETL channels become increasingly decorrelated with depth as they specialise in different local directions. Sparsity remains in a medium range for all three layer types, but ETL tends to be more sparse at intermediate depths, concentrating activations on a subset of salient rays, while LBi-ML and CAL keep weak but widespread responses that preserve the global scaffold.

Taken together, in the shallowest block the ETL layers act as the primary high-frequency extractors, capturing fine-scale edges and textures, whereas LBi-ML layers build a smooth low-frequency backbone and CAL layers modulate this backbone via cross-attention.

**Effective receptive fields at Block 1.** Fig. 13 visualizes normalized effective receptive fields (ERFs) for all layers in Block 0 on an image with strong radial structures. The top row corresponds to the global-context path, where even-indexed layers are LBi-ML and odd-indexed layers are CAL. The bottom row shows ERFs of the local-detail path composed of ETL.

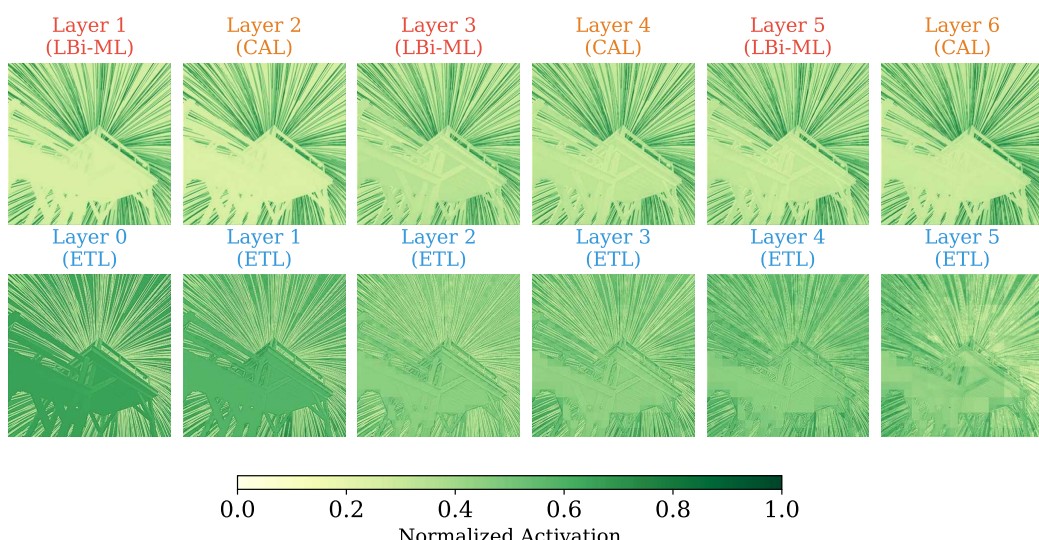

Figure 13: Block 1 ERF visualization for LBi-ML, CAL, and ETL.

The ERFs of LBi-ML layers are broad and diffuse. Activations spread along entire rays from the centre towards the image boundary with relatively uniform intensity, and there is little evidence of sharp peaks. This pattern indicates that LBi-ML aggregates information over long, directional trajectories and encodes global geometric structure rather than isolated local edges. The wide support and smooth profiles are consistent with the state-space formulation, where evidence is propagated along the scan direction to build a coherent global scaffold.

In CAL layers inherit the wide support of the preceding LBi-ML ERFs but introduce mild reweighting. Their ERFs slightly accentuate intersections around the tower and certain beam crossings, while maintaining a generally smooth response elsewhere. Compared with LBi-ML, CAL does not create new localized peaks; instead, it acts as a modulation mechanism that redistributes global-context features conditioned on the information coming from the other branch. This behaviour supports the view of CAL as a cross-branch gating layer rather than an additional high-frequency extractor.

In contrast, ETL ERFs are more concentrated and anisotropic. High responses tightly follow the radial beams and tower edges, and the background is strongly suppressed, producing sharper and more localized patterns than those of LBi-ML or CAL. Across depth, ETL ERFs become increasingly selective around salient lines and corners, demonstrating that the local-detail path focuses on fine-scale structures and high-frequency content already in Block 0.

The ERF shapes reveal a clear division of labour at the shallow stage, LBi-ML layers construct a smooth, globally coherent scaffold, CAL layers modulate this scaffold through cross-branch attention with only mild localization, and ETL layers provide sharply peaked, edge-aligned ERFs. This supports our design intuition that high-frequency details are initially captured by the Transformer-based local path, while the Mamba-based global path concentrates on long-range geometry, preparing the ground for deeper blocks where CAL progressively injects detailed evidence into the Mamba states.

**Mid–shallow behaviour.** Fig. 14 examines Block 2, where features from the first ETB and MCB have already undergone one round of interaction. The ETL layers in the local-detail path still produce sharp, edge-aligned responses along the beams and tower structure, but the activations become more structured and slightly sparser than in Block 1, concentrating on structurally important rays rather than all high-contrast pixels. In the global-context path, the LBi-ML layers now exhibit notice-

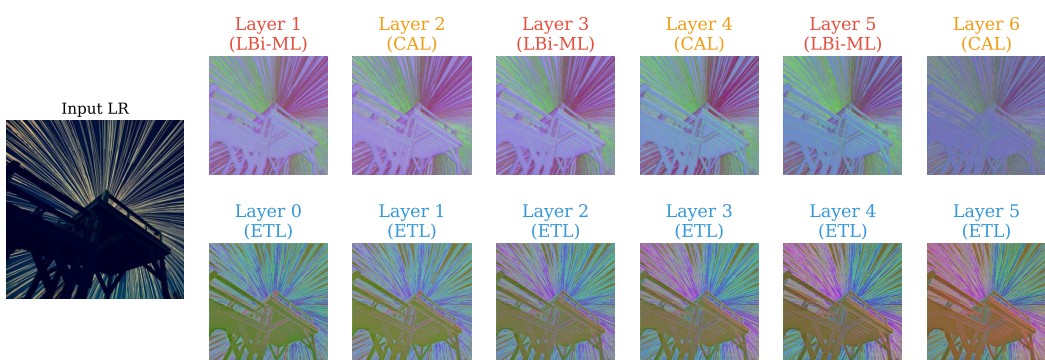

Figure 14: Comparison more details between second ETB and MCB.

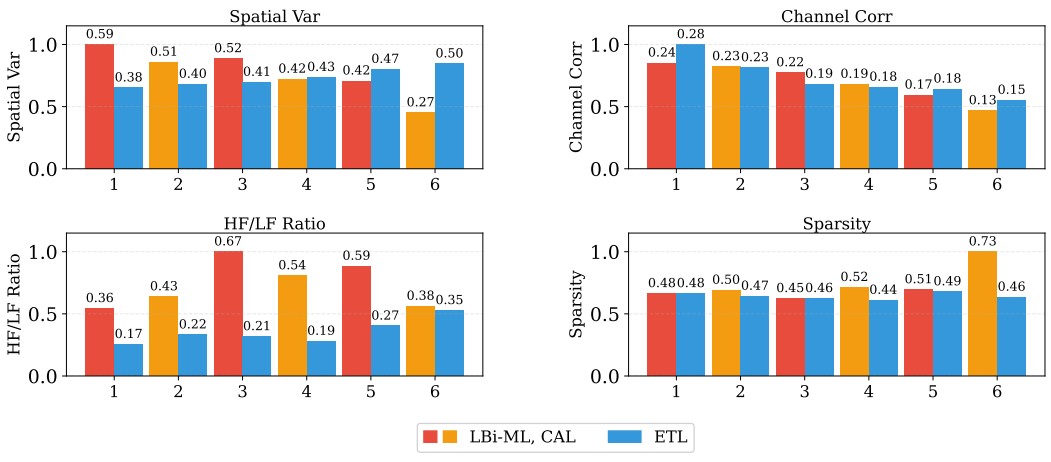

Figure 15: Normalized quantitative statistics for the second ETB and MCB.

ably higher contrast than in the shallowest block: responses follow the main directions of the beams and begin to emphasise the tower silhouette, indicating that the Mamba stream has started to absorb high-frequency cues. The CAL layers sit between successive LBi-ML layers and further refine these patterns, compared with the preceding LBi-ML outputs, CAL feature maps more strongly highlight beam intersections and edges around the tower while attenuating less relevant regions, evidencing cross-branch modulation driven by information coming from ETL.

The quantitative statistics in Fig. 15 support this interpretation. Spatial variance in LBi-ML and CAL becomes comparable to, and in several layers slightly exceeds, that of ETL, showing that the global-context path now contributes substantially to spatial contrast. The HF/LF ratios of LBi-ML layers rise significantly compared with Block 1 and are higher than those of ETL at most depths, indicating that high-frequency energy has been transferred into and amplified within the Mamba stream. CAL layers typically exhibit HF/LF values between those of the surrounding LBi-ML layers and ETL: they neither suppress high-frequency content as strongly as in Block 1 nor dominate it, suggesting

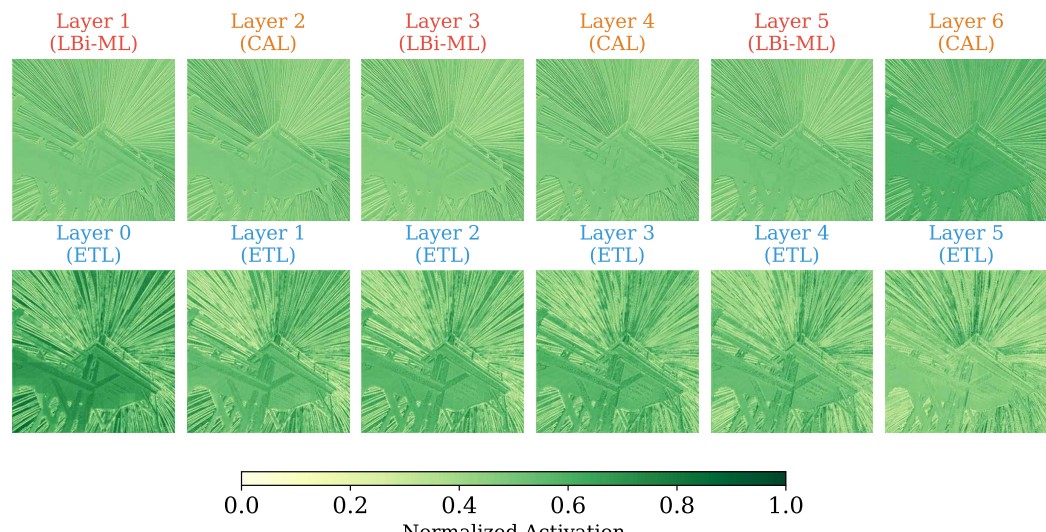

Figure 16: Block 2 ERF visualization for LBi-ML, CAL, and ETL.

a gating role that selects which ETL-derived details should pass into the next LBi-ML. Channel correlation in the global-context path remains moderately high but decreases with depth, reflecting a gradual diversification of Mamba channels, while ETL channels continue to decorrelate as they specialise in distinct local patterns. Sparsity stays in a medium range for all three layer types; LBi-ML becomes more sparse than in Block 1, and CAL typically exhibits the highest sparsity within the global path, meaning it passes only a compact set of important responses to the following LBi-ML. Taken together, Block 2 marks the stage where the Mamba-based path begins to share and partially take over the high-frequency modelling previously dominated by ETL, with CAL acting as a selective cross-attention gate between the two streams.

**Effective receptive fields at Block 2.** Fig. 16 shows the layer-wise ERFs in Block 2. Compared with Block 1, LBi-ML ERFs become more anisotropic: in the first LBi-ML of this block, strong responses already track the dominant beams while still covering the full image, deeper LBi-ML layers further strengthen the tower outline and a subset of structurally important rays, while background regions are gradually attenuated. This indicates that the Mamba states now encode both long-range geometry and non-negligible high-frequency detail.

CAL layers in Block 2 inherit the broad spatial support of the preceding LBi-ML ERFs but display more pronounced, localised peaks at beam crossings and around the tower apex. From the first to the last CAL within the block, these peaks become sharper and the surrounding regions become weaker, revealing that CAL progressively focuses the global scaffold onto a few key locations. This behaviour is consistent with its role as a cross-branch attention module, CAL draws detailed evidence from the ETL path and injects it into the Mamba stream at geometrically meaningful positions.

The ETL ERFs in Block 2 are more concentrated than in Block 1. Early ETL layers still respond along many beams, but deeper ETLs restrict their support to a sparse set of dominant directions and corners, with the background largely suppressed. Thus, while ETL continues to act as a high-frequency extractor, its ERFs show that it now provides a thinner, more selective set of cues, which CAL subsequently routes into the LBi-ML layers. Overall, Block 2 ERFs reveal the beginning of a shift, ETL supplies refined local details, CAL filters and aligns them with the global geometry, and LBi-ML starts to carry a detailed yet coherent representation that prepares the network for the mid–deep and deep blocks.

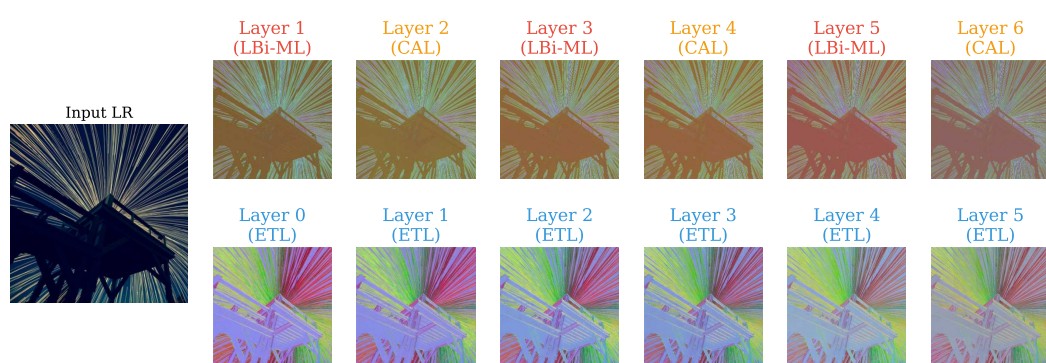

Figure 17: Comparison more details between third ETB and MCB.

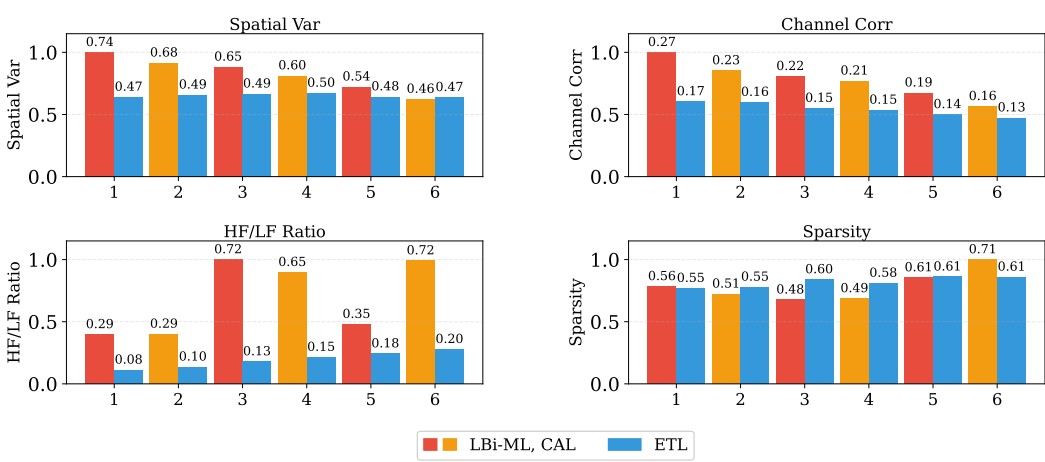

Figure 18: Normalized quantitative statistics for third ETB and MCB).

**Mid–deep behaviour.** Fig. 17 analyzes the third block, where features have passed through two rounds of ETB–MCB interaction. Compared with Block 2, the ETL features in the local-detail path become noticeably more structured and sparse: in early ETL layers, activations still follow many radial beams, but deeper ETLs concentrate almost exclusively on the dominant directions and corners of the tower, indicating that the Transformer branch now provides a compact set of high-frequency cues rather than a dense edge map. In the global-context path, the LBi-ML layers at Block 3 exhibit much stronger contrast than in the preceding block; their feature maps sharply highlight the tower silhouette and several principal beams across the entire image, demonstrating that the Mamba stream has evolved from a coarse scaffold into a detailed, yet globally coherent representation. CAL layers sit between these LBi-ML layers and further refine the global pattern, relative to the preceding LBi-ML outputs, CAL features enhance beam crossings and tower boundaries while attenuating less relevant regions, showing that cross-branch attention is continually re-aligning Mamba states with ETL-derived structure.

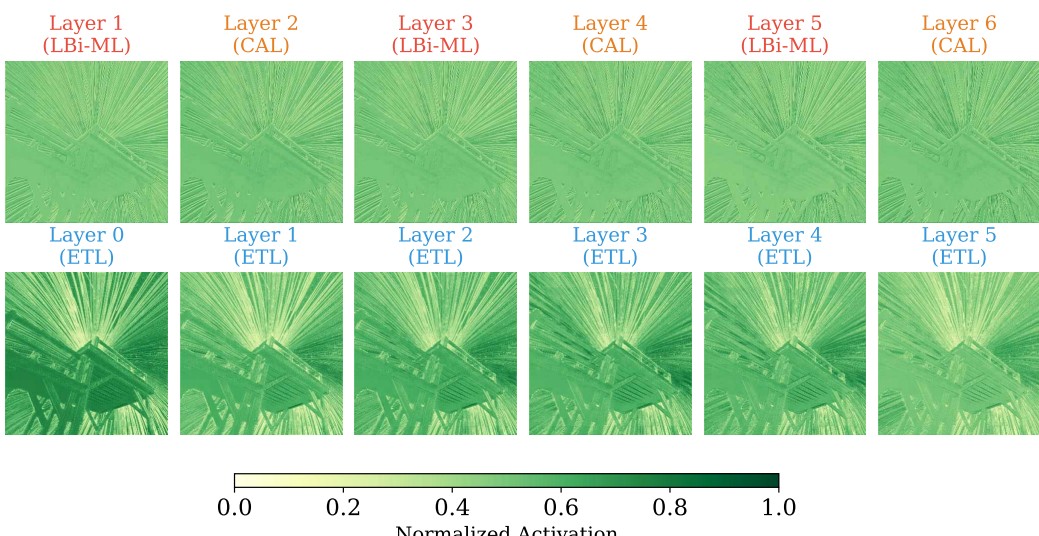

Figure 19: Block 3 ERF visualization for LBi-ML, CAL, and ETL.

The quantitative trends in Fig. 18 corroborate this picture. For spatial variance, the global-context path (LBi-ML/CAL) achieves higher values than ETL in most layers, indicating that a large portion of spatial contrast is now carried by the Mamba stream. The HF/LF ratios of LBi-ML and CAL are also consistently high, with a sharp peak at Layer 5, while ETL reaches similarly high values only at a few depths and remains lower elsewhere. This suggests that high-frequency energy is no longer concentrated solely in the local branch: the Mamba-based path has become a major, and in some layers dominant, high-frequency carrier. Channel correlation remains slightly larger for LBi-ML/CAL than for ETL, indicating globally coherent but increasingly diversified channels in the Mamba stream, while ETL channels continue to decorrelate as they specialise in distinct local patterns. Sparsity is moderate for all three layer types, but ETL shows higher sparsity in deeper layers, whereas LBi-ML/CAL maintain dense yet focused activations, consistent with the idea that the global path now stores rich, high-energy details and the local path provides a thinner structural mask.

CAL's statistics fall between those of LBi-ML and ETL. Its spatial variance and HF/LF ratio are typically lower than the adjacent LBi-ML layer but higher than ETL at the same depth, while its sparsity is often the largest within the global path. Together, these trends support the view of CAL as a selective cross-attention gate, it filters a sparse subset of ETL's high-frequency responses and routes them into the next LBi-ML, where they are amplified and propagated along long-range Mamba trajectories.

**Effective receptive fields at Block 3.** Fig. 19 provides a layer-wise view of the ERFs in Block 3. The first LBi-ML in this block exhibits a broad but clearly anisotropic ERF, strong responses align with the main beams and the tower contour, yet the support still covers almost the entire image. The second and third LBi-ML layers further sharpen these responses, their ERFs emphasise a subset of structurally critical beams and the tower silhouette while progressively suppressing background, showing that Mamba states now encode both long-range geometry and substantial high-frequency detail.

CAL layers inherit the wide spatial support of the preceding LBi-ML ERFs but introduce more localised peaks around beam crossings and the tower apex. From the first to the last CAL in Block 3, these peaks become sharper and the surrounding regions weaker, revealing an increasingly selective

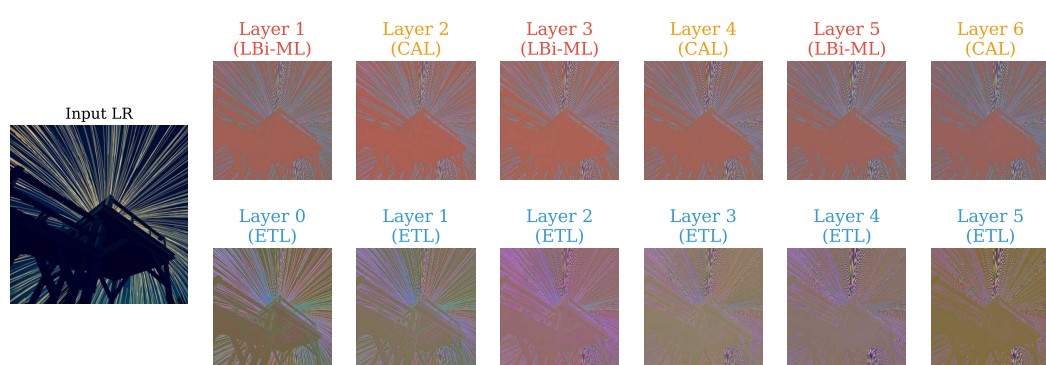

Figure 20: Comparison more details between fourth ETB and MCB.

focus on geometrically important locations. This behaviour is precisely what our design intends: CAL draws detailed evidence from the ETL branch and injects it into the Mamba stream only where it aligns with the global scaffold.

The ETL ERFs in Block 3 are tighter and more selective than in Block 2. Early ETLs still respond along several beams, but deeper ETLs restrict their support to a few dominant rays and corners, with the rest of the image nearly inactive. Thus, ETL at this stage supplies a compact, edge-aligned signal that CAL can gate into LBi-ML. Overall, Block 3 ERFs show that the network has reached a mid–deep regime in which ETL acts as a sparse provider of structural cues, CAL as a geometry-aware filter, and LBi-ML as the main carrier of globally consistent high-frequency details.

**Deep-layer behaviour.** Fig. 20 analyzes the deepest block, after three rounds of ETB–MCB interaction. In the local-detail path, ETL feature maps have become extremely sparse and structured, early ETL layers still respond along several beams, but deeper ETLs concentrate almost exclusively on the tower silhouette and a few dominant directions, leaving most of the image nearly inactive. This indicates that, by Block 4, the Transformer branch mainly provides a thin structural mask and no longer carries dense high-frequency content. In contrast, the LBi-ML layers in the global-context path now exhibit very strong contrast across the entire image, their feature maps sharply highlight the tower outline and principal beams while suppressing background, showing that the Mamba stream has evolved into the primary carrier of detailed but globally coherent information. CAL layers sit between successive LBi-ML layers and refine these patterns, compared with the preceding LBi-ML outputs, CAL features further emphasise beam crossings and tower boundaries and downweight less relevant regions, revealing that cross-branch attention is still actively shaping where Mamba allocates its high-frequency energy at the deepest stage.

The quantitative statistics in Fig. 21 make this shift explicit. In terms of spatial variance, the global-context path strongly dominates ETL at all depths, its normalized variance grows from $1.02$ to $1.73$ from Layer 1 to Layer 6, whereas ETL only rises from $0.54$ to $1.23$. Thus, most spatial contrast and edge strength now resides in the Mamba-based stream. The HF/LF ratios show a similar pattern, LBi-ML/CAL layers maintain a pronounced high-frequency bias, while ETL stays in a low band , indicating that the Transformer path encodes mainly low- and mid-frequency structure at this stage. Channel correlation for the global path lies between $0.21$ and $0.18$, consistently higher than ETL ($0.17$ to $0.12$), suggesting globally coherent but diversified Mamba channels, while ETL channels are more decorrelated as they specialise in a few structural cues. Sparsity is high for both paths, yet ETL tends to be slightly more sparse, whereas LBi-ML/CAL maintain somewhat denser activations,

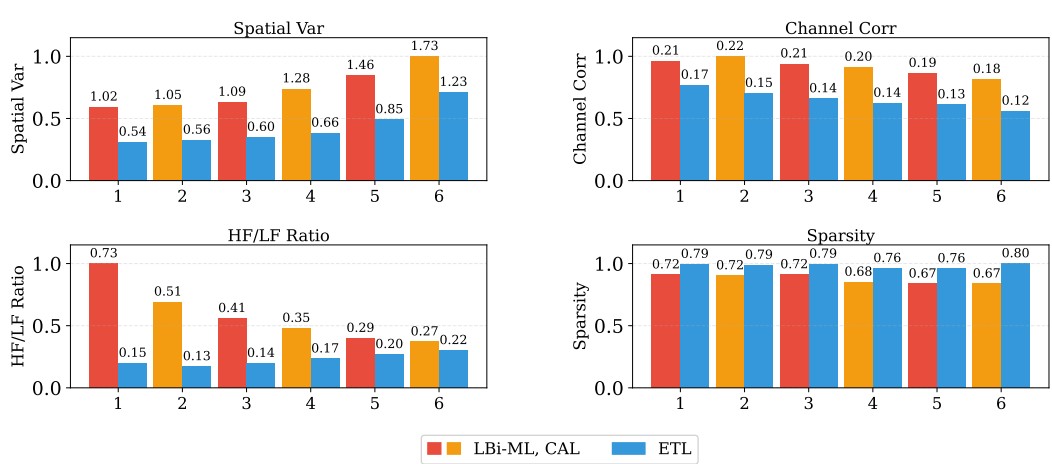

Figure 21: Normalized quantitative statistics for Block 4.

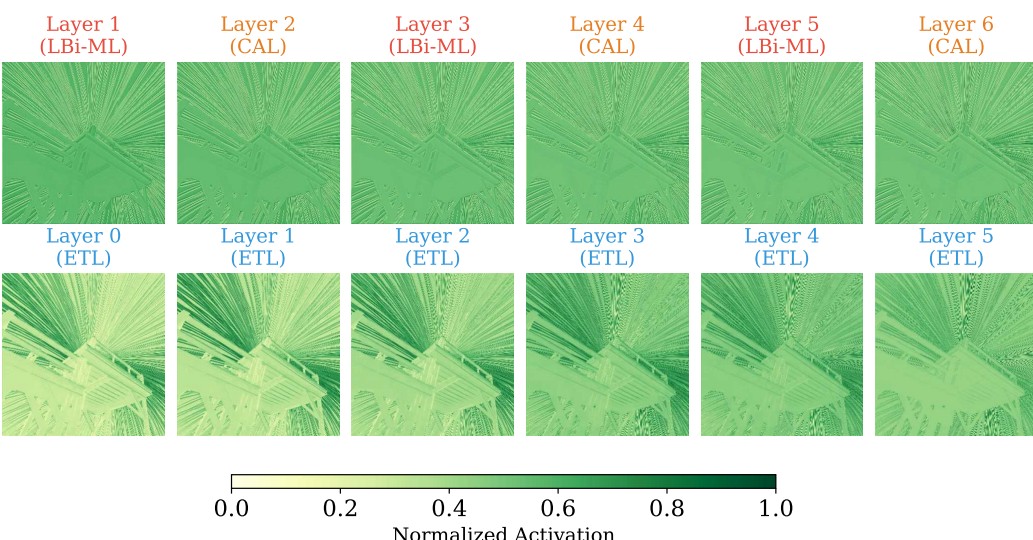

Figure 22: Block 4 ERF visualization for LBi-ML, CAL, and ETL.

consistent with the picture that the local branch has collapsed to a very selective structural mask, while the global path retains rich, high-energy detail.

Within this block, CAL's statistics fall between those of LBi-ML and ETL. Its spatial variance and HF/LF ratio are systematically lower than in the adjacent LBi-ML layers but higher than in ETL, and its sparsity is often the largest within the global path. This combination, moderate variance, relatively high HF/LF, and high sparsity, supports the interpretation of CAL as a deep, highly selective cross-attention gate, CAL picks a compact set of high-frequency signals from the ETL branch and injects them into the next LBi-ML at carefully chosen locations, preventing over-sharpening while preserving global consistency in the Mamba representation.

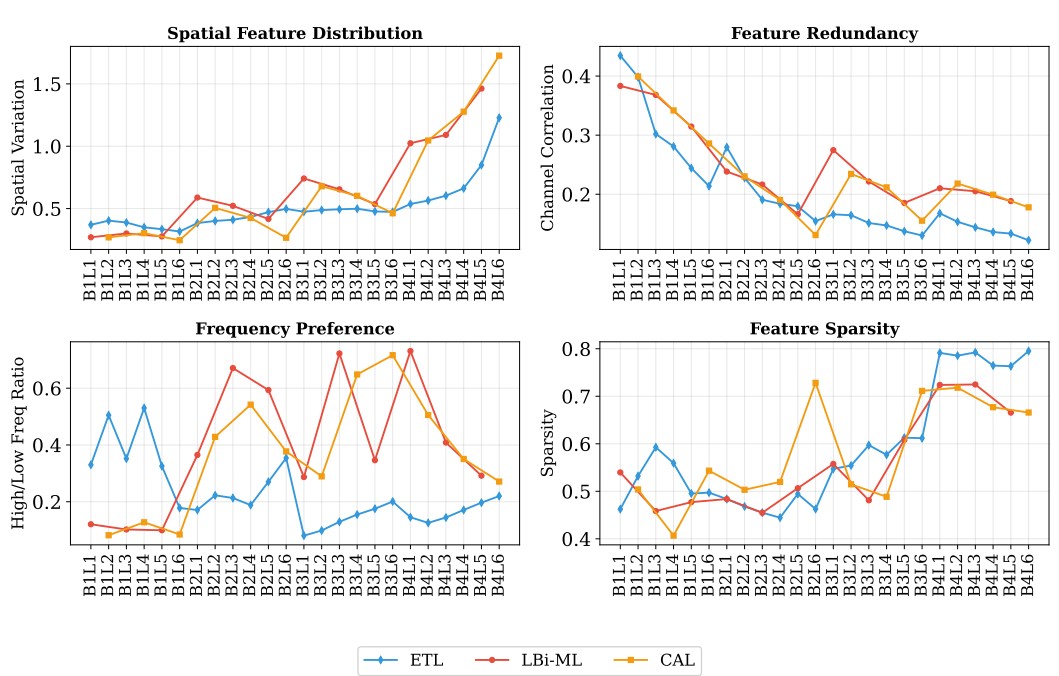

Figure 23: Overall layer-wise comparison across all blocks.

**Effective receptive fields at Block 4.** Fig. 22 provides a layer-wise view of the ERFs in Block 4. The first LBi-ML layer in this block exhibits a broad yet strongly anisotropic ERF, high responses align with the main beams and the tower contour across the full image, showing that Mamba states simultaneously encode long-range geometry and strong edge information. Deeper LBi-ML layers further sharpen these patterns; their ERFs emphasise a subset of structurally critical beams and the tower silhouette while progressively suppressing the background, yielding a globally coherent but highly focused representation.

CAL ERFs inherit the wide spatial support of the preceding LBi-ML but exhibit even more pronounced local peaks around beam crossings, corners, and the tower apex. From the first to the last CAL layer in Block 4, these peaks become sharper and the surrounding regions weaker, indicating that CAL concentrates the global scaffold onto a small set of geometrically meaningful sites. This behaviour is precisely in line with its intended role, CAL filters high-frequency evidence coming from ETL and injects it into the Mamba stream only where it aligns with the global structure learned by LBi-ML.

The ETL ERFs at Block 4 are tighter and more selective than in the preceding blocks. Early ETL layers still respond along several beams, but the deepest ETLs restrict their support to a few dominant rays and to the tower's key edges, with most other locations nearly inactctive. Thus, in the final block, ETL acts as a sparse provider of structural cues. CAL, with geometry-aware attention, decides which of these cues are forwarded and LBi-ML serves as the main carrier of a dense, globally consistent, high-frequency representation. Together, these ERFs confirm that by Block 4 the network has reached a mature division of labour: the Transformer path supplies a compact structural prior, CAL acts as a deep cross-branch gate, and the Mamba-based global-context path dominates both spatial contrast and high-frequency detail in support of faithful super-resolution.

**Overall layer-wise trends across blocks.** Fig. 23 summarizes the evolution of the four statistics, spatial variance, channel correlation, HF/LF ratio, and sparsity, over all layers from the first to the last block. The three curves correspond to the Transformer-based ETL layers (blue), the Mamba-based LBi-ML layers (red), and the cross-attention CAL layers (yellow), allowing us to track how the local-detail path, global-context path, and cross-branch modulation interact as depth increases.

In terms of spatial variance, the ETL curve exhibits relatively high values in the first block and then stabilizes at a moderate level, reflecting strong edge contrast at shallow layers followed by a gradual consolidation into a thin structural mask. By contrast, the Mamba curve starts with low variance and grows steadily across blocks, eventually overtaking ETL in the mid–deep and deep layers. CAL remains between the two, its variance is slightly above LBi-ML in the earliest layers of each block, where it first reweights global features using local evidence, and then closely follows the Mamba trend. This pattern confirms that spatial contrast is initially concentrated in the Transformer branch and progressively migrates into the Mamba stream as the network deepens.

The HF/LF ratio reveals a similar shift in frequency preference. ETL maintains moderate high-frequency content in early layers, consistent with its role as the initial edge and texture extractor, but its HF/LF values do not increase substantially with depth. In contrast, the Mamba curve rises sharply from the second block onwards and remains high in mid–deep and deep layers, showing that the Mamba-based path becomes the dominant high-frequency carrier. The CAL curve starts low in the first block, where it mostly relays global context, but develops pronounced peaks in later blocks that nearly match the Mamba HF/LF ratio. These peaks indicate that CAL is not merely passing information through, it acts as a frequency-aware gate that injects high-frequency cues from ETL into the Mamba states at selected depths.

Channel correlation decreases with depth for all three curves, reflecting increasing specialisation of channels. Nonetheless, LBi-ML and CAL retain slightly higher correlation than ETL, suggesting that the global-context path encodes more coherent, shared structures across channels, while the local-detail path diversifies more strongly into distinct orientation- and region-specific filters. This is consistent with our interpretation of Mamba as a global scaffold that organises and propagates information received from the Transformer branch.

Sparsity further clarifies the division of labour. The ETL curve shows relatively high sparsity in deeper layers, indicating that the Transformer branch collapses to a compact structural prior that is active only on a small subset of beams and contours. The Mamba curve maintains medium sparsity, consistent with dense yet focused activations that carry rich detail over the entire image. The CAL curve often attains the highest sparsity among the three, especially in mid–deep layers where its HF/LF ratio is also high. This combination of high HF/LF and high sparsity supports our design view of CAL as a selective cross-attention module: it picks a small number of high-frequency, structurally aligned cues from ETL and injects them into the Mamba stream, rather than propagating dense patterns.

**Beyond Simple Stacking.** Taken together, the above visualizations demonstrate that our model is not a naïve hybrid that simply stacks Mamba and Transformer layers, but a carefully structured dual-path architecture with an emergent division of labour. Across blocks, feature maps, quantitative statistics, and ERFs consistently show three distinct and complementary roles, (i) ETL layers in the ETB act as the initial high-frequency extractor, producing sharp, edge-aligned responses at shallow depth and gradually condensing into a sparse structural prior in deeper blocks, (ii) LBi-ML layers in the MCB start as low-frequency, globally coherent scaffolds and progressively become the main carriers of high- frequency, high-contrast information as depth increases, and (iii) CAL layers serve as sparse, geometry- and frequency-aware cross-attention gates that selectively route ETL-derived details into the Mamba stream at geometrically meaningful locations, rather than introducing another generic attention block.

Table 12: The impact of different window sizes on $2 \times$ SR in Urban100.

| Method | Params | MACs | Urban100 |
|---|---|---|---|
| Win=(4, 8, 16, 32, 48,64) | 826K | 179.2G | **33.29/0.9382** |
| Win=(8, 8, 8, 8, 8,8) | 826K | 172.8G | 33.16/0.9372 |
| Win=(64, 64, 64, 64, 64, 64) | 827K | 181.3G | 33.19/0.9377 |
| Win=(64, 48, 32, 16, 8,4) | 826K | 179.2G | 33.24/0.9378 |

Table 13: Average runtime of different components in DPMFormer on an RTX 4090.

| Components. | ETB | MCB | IEB | LBi-ML | CAL | ETL |
|---|---|---|---|---|---|---|
| Avg. Speed (ms) | 102.75 | 54.15 | 6.65 | 4.34 | 4.96 | 17.13 |

This depth-dependent behaviour would not arise from a mere juxtaposition of Mamba and Transformer modules with similar functionality. Instead, the visual evidence reveals a coordinated, coarse-to-fine reallocation of modelling capacity: high-frequency content is first captured locally by ETL, then transferred and amplified along long-range Mamba trajectories through CAL, while ETL itself collapses into a thin structural mask. The resulting representation in the deepest blocks is a globally consistent, Mamba-dominated feature field enriched with selectively injected local details. These observations provide a mechanistic justification for our design: the dual-path composition is *reasonable*, each component assumes a clear, complementary role, *effective*, high-frequency modelling shifts to the more efficient Mamba path at depth, in line with our efficiency results, and *innovative*, the CAL-driven transfer of frequency content from a Transformer path into a state-space path is qualitatively different from prior single-path Mamba or Transformer architectures.

### A.13.3 IMPACT OF DIFFERENT WINDOW SIZES

We adopt a variable-window strategy in the ETB to progressively enlarge the local receptive field across depth. To study its impact, we compare several window schedules under the same parameter budget on $\times 2$ SR with Urban100, as reported in Table 12. The default configuration Win=(4, 8, 16, 32, 48, 64) gradually increases the window size from shallow to deep ETL layers and achieves the best accuracy with essentially the same parameters and MACs as the alternatives. Using a small and fixed window (8,8,8,8,8,8) slightly reduces MACs but leads to a clear drop in PSNR and SSIM, suggesting that a too-local receptive field is insufficient for capturing large-scale structures in Urban100. On the other hand, always using a large window (64,64,64,64,64,64) or applying a reversed schedule (64,48,32,16,8,4) partially recovers performance but remains inferior to the progressive setting, while also incurring comparable or higher MACs. These results indicate that gradually expanding the ETB window along depth offers a favourable trade-off between local detail modelling and global context aggregation.

For CAL, we keep a fixed $8 \times 8$ window rather than adopting variable windows, CAL already operates on the global-context path and serves as a cross-branch gating module, so a moderate and constant window size provides sufficient context for effective cross-attention while keeping the computational cost stable.

### A.13.4 RUNTIME OF INDIVIDUAL COMPONENTS

We further benchmark the runtime of each major component in DPMFormer to understand where the computational bottlenecks lie. Table 13 reports the average latency of one forward pass for the ETB, MCB, and IEB, as well as their internal building blocks, LBi-ML, CAL, and ETL, measured on a single NVIDIA RTX 4090 under our default $\times 2$ SR validation setting.

From Table 13, the ETB is roughly $1.9\times$ slower than the MCB, reflecting the fact that windowed Transformer layers remain the dominant cost in the network. At the layer level, a single ETL takes 17.13 ms on average, which is about $3.9\times$ and $3.5\times$ slower than an LBi-ML layer (4.34 ms) and a CAL layer (4.96 ms), respectively. This confirms that the Mamba-based global-context branch is substantially more efficient than the Transformer-based local-detail branch, and justifies our design choice of allocating more depth to the MCB while keeping the number of ETBs moderate.

The IEB introduces only 6.65 ms of overhead, which corresponds to about $4\%$ of the combined ETB+MCB latency. Given the performance gains observed in our ablations, this small runtime cost is well justified: IEB enables effective information flow between the two paths without becoming a computational bottleneck.

Overall, these measurements show that the proposed decomposition into ETL, LBi-ML and CAL yields a favourable efficiency profile. Most computation is spent in a limited number of Transformer blocks, while the majority of depth resides in lightweight Mamba and cross-attention layers, allowing DPMFormer to enjoy strong performance with competitive runtime.

A.14  EFFICIENT ON 2K AND 4K IMAGE

Considering that Mamba-style state space layers scale linearly in both computation and memory with respect to the sequence length, they are particularly attractive for ultra-high-resolution SR. We therefore evaluate $2\times$ upsampling on 2K and 4K inputs, using SRFormer, HiT-SRF, CATANet, MambaIRv2, and our DPMFormer on 100 images for each resolution. The efficiency results are reported in Table 14.

Table 14: Efficiency of different SR models on 2K and 4K $2\times$ super-resolution.

| Method | ImageSize | MACs (G) | FLOPs (G) | Infer. Time (ms) $\uparrow$ | Mem. Alloc. (MB) $\downarrow$ |
|---|---|---|---|---|---|
| SRFormer | 2K | 596.6 | 1139.1 | 5280 | 134.90 |
| HiT-SRF | 2K | 551.3 | 1102.6 | **1235** | 119.33 |
| CATANet | 2K | | | Out of Memory (OOM) | |
| MambaIRv2 | 2K | 646.8 | 1293.6 | 2856 | 120.21 |
| **DPMFormer** | 2K | 449.5 | 898.9 | 1556 | **43.86** |
| SRFormer | 4K | 2278.3 | 4556.5 | 17048 | 459.89 |
| HiT-SRF | 4K | 2186.3 | 4372.6 | 17668 | 442.06 |
| CATANet | 4K | | | Out of Memory (OOM) | |
| MambaIRv2 | 4K | 2587.1 | 5174.3 | 28099 | 445.20 |
| **DPMFormer** | 4K | 1785.7 | 3571.4 | **6805** | **139.47** |

From Table 14, we observe that DPMFormer consistently achieves the best trade-off between computation, runtime, and memory footprint at both resolutions. On 2K inputs, DPMFormer reduces MACs and FLOPs by roughly 18–30% compared with SRFormer, HiT-SRF, and MambaIRv2, while requiring only 43.86 MB of GPU memory, which is about $2.7\times$–$3.1\times$ smaller than the transformer and Mamba baselines. Its inference time (1556 ms) is comparable to the high-throughput HiT-SRF (1235 ms), and substantially faster than SRFormer and MambaIRv2 (about $3.4\times$ and $1.8\times$ speed-ups, respectively).

When scaling to 4K images, the advantages become more pronounced. DPMFormer maintains 20–31% fewer MACs than the baselines and achieves the lowest memory usage (139.47 MB), yielding about a $3\times$ reduction over SRFormer, HiT-SRF, and MambaIRv2. In terms of runtime, DPMFormer runs $2.5$–$4.1\times$ faster than these models on 4K inputs, while CATANet runs out of memory at both 2K and 4K resolutions. These results confirm that the proposed dual-path Mamba–Transformer design scales more gracefully to ultra-high-resolution SR than purely transformer-based or Mamba-only architectures.

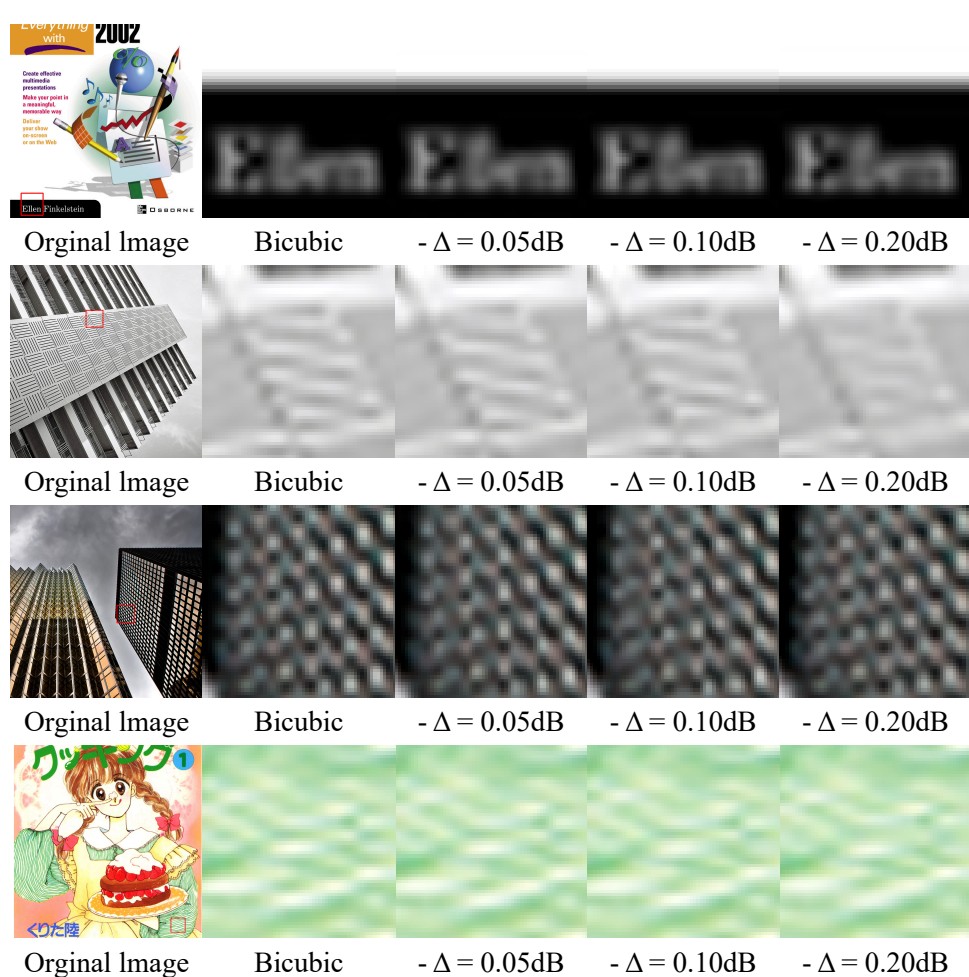

Figure 24: Visual impact of small PSNR differences.

### A.15 PERCEPTUAL IMPACT OF SMALL PSNR CHANGES.

To put the PSNR margins reported in Section 4.3 into context, Fig. 24 illustrates how small PSNR variations translate into visible differences on several high–frequency patterns. For each row, the leftmost image shows the original HR image with a red ROI, the second column shows the bicubic $4\times$ upsampled reference patch, and the remaining three columns apply progressively stronger mild blurring such that the PSNR of the patch decreases by $\Delta 0.05$ dB, $\Delta 0.10$ dB, and $\Delta 0.20$ dB, respectively. Here the symbol $\Delta$ denotes the PSNR change with respect to the same bicubic reference. Across all four examples, even a reduction of $\Delta 0.05$ dB already produces a visible loss of sharpness when inspected at normal zoom. In the first row (text), the character edges become slightly fuzzy and the stroke contrast decreases. In the second and third rows (building façades and grid structures), the fine grooves and checkerboard-like patterns lose crispness and local contrast, and the regular lines begin to merge. In the fourth row (cartoon-like texture), the wavy details gradually flatten out. At $\Delta 0.10$ dB these effects become clearly noticeable: line structures are smeared, repeated patterns appear visibly washed out, and thin strokes in the text example start to collapse. By $\Delta 0.20$ dB, most

of the high–frequency detail in the ROIs is largely destroyed and the patches look almost uniformly blurred.

These examples show that PSNR differences in the range of 0.05–0.10 dB, especially on highly textured or structured regions, correspond to perceptible quality changes rather than pure numerical noise. This supports the relevance of the 0.05–0.10 dB gains reported by DPMFormer on challenging datasets such as Urban100 and Manga109.

ANONYMOUS CODE

For double-blind review, we provide an *anonymized GitHub repository* at `https://anonymous.4open.science/r/DPMFormer-BAA4`, containing everything needed to reproduce the paper's results: source code, training/evaluation scripts, configuration files (covering window sizes, ETB counts, per-stage channel widths and the ACLinear schedule $C_1 \leftrightarrow C_2$, CAL window size, DW-SwiFFN ratio, and LBi-Mamba settings), data preparation utilities. Upon acceptance, we will de-anonymize the repository, add a license and long-term DOI/archival tag, and keep the code and artifacts publicly available.

LARGE LANGUAGE MODEL USAGE STATEMENT

We used a large language model *only for language editing* (grammar/typo correction, style tightening, and minor phrasing for clarity). The LLM was *not* used to generate algorithms, experimental designs, analyses, figures, tables, code, or claims. All technical content (methods, proofs/derivations, experiments, metrics, and conclusions) was authored and validated by the authors, and every edited sentence was reviewed by a human author for accuracy and scope. No proprietary or personally identifiable data were provided to the LLM.

