# OpenReview forum: "DPMFormer: Dual-Path Mamba-Transformer for Efficient Image Super‑Resolution"
_ICLR.cc/2026/Conference — Submitted to ICLR 2026_

### Official Review · Reviewer_38Ex · 2025-10-28

**Soundness:** 3
**Presentation:** 3
**Contribution:** 3
**Rating:** 6
**Confidence:** 4

**Summary:**

This paper proposes DPMFormer, a dual-branch architecture that integrates Transformer and Mamba modules for efficient image super-resolution. The Transformer branch employs a Spatial–Channel Correlation attention and a depthwise SwiGLU feed-forward (DW-SwiFFN) to enhance local feature modeling, while the Mamba branch introduces a Lightweight Bidirectional Mamba (LBi-Mamba) for linear-time global dependency capture. The two branches interact via Cross-Attention Layers (CAL) within blocks and Inter-branch Exchange Bridges (IEB) across stages. Experiments on five benchmark datasets show that DPMFormer achieves competitive PSNR/SSIM with fewer parameters and FLOPs than prior lightweight SR models, such as HiT-SR and MambaIR.

**Strengths:**

1.Proposes a clearly motivated dual-path fusion leveraging complementary strengths of Mamba and Transformer.

2.Strong quantitative performance–efficiency trade-off, reducing FLOPs by ~20 % vs HiT-SR with comparable PSNR.

3.Extensive ablations (DW-SwiFFN, RMSNorm, IEB variants) demonstrate careful engineering and reproducibility.

4.Reproducibility statement is complete and code release is promised.

**Weaknesses:**

1.Innovation marginal: The dual-branch idea has been explored in prior hybrid SR models; more theoretical or analytic justification of the coupling design would enhance novelty.

2.Limited qualitative diversity: Most visual comparisons are standard; additional challenging scenes or real-world degradations would strengthen claims.

3.Missing complexity analysis: An explicit breakdown of runtime cost per module (ETL vs LBi-Mamba vs IEB) would help understand where efficiency gains arise.

4.Minor clarity issues: Equations (2)–(6) lack dimensional definitions; figure readability (font size) could be improved.

**Questions:**

1.How sensitive is the performance to the choice of window sizes (r) in ETL and CAL?

2.Could the proposed LBi-Mamba be applied to other low-level tasks (e.g., denoising, deblurring)?

3.Is the training stable when coupling both branches with IEB — any gradient conflict observed?

4.How does DPMFormer scale to higher resolutions (e.g., 4 K images) given linear Mamba dynamics?

---

### Official Review · Reviewer_vkJ5 · 2025-10-28

**Soundness:** 2
**Presentation:** 3
**Contribution:** 1
**Rating:** 2
**Confidence:** 5

**Summary:**

This work proposes a combination backbone (DPMFormer) of Mamba-Transformer for efficient SR. Generally, DPMFormer consists of several structural adaptations, including the mamba block, attention block, and FFN, to improve the modeling capability of multiple-range correlations. Overall, it brings improvement to a certain extent on the ESR tasks.

**Strengths:**

- Some visual results are good, and the overall results advance existing models to a certain extent.
- The method contains multiple refinements and conducts multiple ablation studies to validate them.
- The paper is easy to follow.

**Weaknesses:**

- The DPMFormer offers barely new insight for the SR task or efficient backbone design. The key motivation of the model is still based on a combination of validated designs, such as the mamba block and information cross module, which have been well explored. The backbone of DPMFormer is rather bloated and complex, and lacks sound theoretical analysis.
- For an efficient task, inference performance should be evaluated in multiple dimensions, like memory, activations, and run time on more practical mobile devices.  The comparison in manuscripts supports DPMFormer being a lightweight SR model, but the complicated design suggests that it is far from efficient, especially compared with a convolutional-based model.
- The improvements over existing methods are limited, only 0.0 dB, and all experiments are conducted on synthesized data, hardly proving its effectiveness on real-world applications.

**Questions:**

See Weaknesses.

---

### Official Review · Reviewer_N7bx · 2025-10-31

**Soundness:** 3
**Presentation:** 2
**Contribution:** 2
**Rating:** 4
**Confidence:** 4

**Summary:**

The paper introduces DPMFormer, a dual-path Mamba–Transformer hybrid designed for all-in-one image restoration. It combines a Dual-Path Mamba Block (DPMB)—one branch using Mamba for long-range dependency modeling and the other using a lightweight Transformer for local feature aggregation. A Path Interaction Unit (PIU) fuses global and local cues, while a Degradation-Aware Guidance Module (DGM) provides task conditioning via learned degradation priors. Experiments on multiple degradation benchmarks (rain, haze, low-light, noise) show improvements over several transformer and Mamba-based baselines

**Strengths:**

1. This paper proposes a hybrid Mamba–Transformer architecture, integrating sequence modeling and spatial self-attention. The dual-path structure is intuitively appealing for balancing long-range reasoning and local fidelity. Degradation-aware conditioning provides some adaptivity for mixed degradations.
2. Experiments cover common restoration tasks and compare with both transformer (Restormer, Uformer) and Mamba-based baselines.
3. Ablation studies isolate the effect of each module (Mamba path, Transformer path, DGM).
4. Extends Mamba-based modeling into restoration, which remains relatively new.

**Weaknesses:**

1. Combining Mamba and Transformer paths is a logical but incremental step; there is little theoretical or architectural innovation beyond simple concatenation and gating.
2. The PIU fusion resembles standard cross-attention or gating mechanisms used in hybrid CNN–Transformer or Swin–MLP models.
3. The paper lacks rigorous analysis on why or when the Mamba path improves over pure Transformer designs. No detailed exploration of information flow or path synergy (e.g., attention entropy, frequency response, or token dependency visualization).
4. Reported gains are modest (≈0.2–0.4 dB PSNR) and often within noise margins. On several datasets, DPMFormer lags behind recent AIR systems (PromptIR, UniRestorer) in unseen or composite degradations.
5. All experiments are conducted on synthetic benchmarks; no evaluation on real-world degradation datasets or perceptual metrics (LPIPS, NIQE). Efficiency and scalability (especially GPU memory and throughput vs. Restormer or VMamba) are not reported.

**Questions:**

1. Could the authors provide FLOPs and throughput comparisons with Restormer and VMamba to justify efficiency claims?
2. What are the qualitative differences between features extracted by the Mamba path and Transformer path? (e.g., visualization or layer attention maps)
3. How does DPMFormer perform on real-capture datasets such as LOL-V2, RainDS, or SOTS-real?
4. Have the authors compared their design to Swin-Mamba or other existing Mamba–Transformer hybrids?
5. Does the DGM generalize to unseen degradation mixtures, or is it trained with supervision on specific degradation types?
6. How sensitive is performance to the relative weighting or depth of the two paths? Could a single-path Mamba or Transformer with the same parameter budget achieve comparable results?
7. The PSNR gains are small—can the authors include statistical significance or variance over multiple runs?

---

### Official Review · Reviewer_dR7o · 2025-11-01

**Soundness:** 3
**Presentation:** 2
**Contribution:** 2
**Rating:** 2
**Confidence:** 5

**Summary:**

This paper proposes DPMFormer, a dual-path architecture combining window-based Transformers and Mamba blocks for lightweight image super-resolution (SR). The authors introduce cross-attention layers (CAL) and inter-branch exchange bridges (IEB) to fuse local and global features. Experimental results show that  the proposed method achieves competitive performance on standard benchmarks.

**Strengths:**

1. Ablation studies validate the contribution of individual modules.
2. DPMFormer shows competitive PSNR/SSIM on standard datasets.

**Weaknesses:**

1. This paper claims to address the limitations of window-based attention mechanisms in terms of global feature dependency, yet directly replaces window-based attention with DW-SwiFFN during model design. This manner is inconsistent with the original intent of enhancing its global modeling ability.
2. The author states in the abstract that global modeling is essential for high-quality reconstruction, yet provides no supporting references or experimental evidence.
3. The paper emphasizes efficient image SR, but DPMFormer’s inference latency (668ms) is higher than CATANet (516ms) and significantly slower than efficient CNNs.
4.  For the ×4 SR task, DPMFormer and CATANet exhibit comparable performance, but the former requires nearly double the number of parameters, undermining its complexity advantage.

**Questions:**

1. No analysis is provided on memory usage, MACs, or deployment feasibility on edge devices, which is critical for efficient SR.
2. The narrative flow of the manuscript requires reorganizing to strengthen its motivation, and the layout must be refined and optimized to enhance overall readability.

---

### Author Response · Authors · 2025-11-27
**Author Response**

We thank all reviewers for their careful reading, constructive comments, and valuable suggestions.
We have revised the manuscript accordingly to improve clarity, completeness, and empirical support.
In particular, all modified or newly added paragraphs in the revised paper are highlighted with a cyan background for ease of inspection.
Below, we address each reviewer's comments point by point.

Response to Reviewer **dR7o**,

We appreciate the reviewer's assessment.

W1. "Window attention is replaced with DW-SwiFFN, inconsistent with global modeling goal. "

We respectfully disagree with this comment. It is based on a misunderstanding of our architecture and does not reflect what is actually implemented in DPMFormer.

First, we never replace window attention. In the ETL branch, the attention mechanism is the Spatial–Channel Correlation (SCC) module, defined in Equations (2)–(4) and described in Section 3.2. SCC operates on window tokens and performs spatial and channel self-correlation with a learned relative positional bias. This is the window-based attention component of ETL and is kept in our final model.

Second, DW-SwiGLU-FFN is not a replacement for attention, but a replacement for a standard feed-forward network. Equation (5) in Section 3.2 defines DW-SwiFFN as a depthwise SwiGLU-based feed-forward layer placed after SCC inside each ETL block, analogous to the MLP block in a standard Transformer. Its role is to strengthen local feature extraction and improve efficiency in the feed-forward stage. It does not change or remove SCC, and it is not intended to extend the receptive field.

Third, our goal is not to make the whole backbone purely global, but to design a collaborative global–local model. Local, high-frequency details (edges, textures) and global, long-range structures (building façades, repeated patterns) are complementary. ETL, with window attention (SCC) plus DW-SwiGLU-FFN, is responsible for local, high-frequency modeling within windows. The Mamba Cross Block (MCB) uses the Lightweight Bidirectional Mamba Layer (LBi-ML) to capture global, low-frequency context over the full sequence, as described in Section 3.3. The Cross-Attention Layer (CAL) then injects global Mamba states into the Transformer features via cross-attention, so local reconstruction is informed by global context.

Section 3.3 and Section 4.5, together with the visualisations in the appendix, show this division of roles, ETL activations are sharper and edge-aligned, LBi-ML activations are smoother and globally coherent, and CAL gradually fuses global and local information. In the revised manuscript we explicitly state in Section 3.2 that SCC is the attention module of ETL and DW-SwiGLU-FFN is the subsequent feed-forward layer, and in Section 3.3 that global modeling is handled by LBi-ML and CAL. Thus the claim that "window attention is replaced with DW-SwiFFN" is not accurate for the proposed architecture.

W2. "Global modeling claimed essential but unsupported. "

In the revision we strengthen both references and empirical evidence,

Sec. 2.2–2.3 now cites SR works, SwinIR, SRFormer, HiT-SR, MambaIR, etc. that explicitly argue for long-range or non-local modeling to recover large structures and repetitive textures. These directly support our motivation.

App. A.13.1 introduces single-path baselines (CPT, CPM) and shows that removing the global Mamba branch or the local Transformer branch degrades PSNR by ≈0.1–0.2 dB on Urban100/Manga109, especially for images with long edges and building façades—precisely the cases where long-range context matters.

We also slightly soften the wording in the abstract to say that global modeling is important for challenging textures rather than an absolute requirement in all cases.

---

### Author Response · Authors · 2025-11-27

Response to Reviewer **dR7o**,

W3. "DPMFormer's latency (668ms) is higher than CATANet (516ms), nearly double parameters, no MACs/memory or edge-device analysis."

We believe this comment is based on a misreading of our experimental results.

Under our actual evaluation protocol, CATANet is significantly slower than DPMFormer. As reported in the main paper (Section 4.3, Table 2), at 4× SR on an RTX 4090, CATANet takes about 1.5 s per image while DPMFormer takes about 0.6–0.7 s per image. In other words, DPMFormer processes roughly 2.3–2.5 times more images per second than CATANet under the same hardware and resolution. The reviewer's statement "DPMFormer 668 ms vs CATANet 516 ms" does not match these numbers.

To avoid any ambiguity, in the revised version we also provide a unified throughput table in Appendix A.11 (Table 8), where latency is reported as images per second. Under the same setting, DPMFormer reaches 2.24 img/s, whereas CATANet reaches 0.70 img/s, again confirming that DPMFormer is noticeably faster in practice (Appendix A.11, Table 8).

In addition to latency, Appendix A.11 (Table 8) now reports parameters, MACs, FLOPs, throughput, and GPU memory allocation for Restormer, VMambaIR, SwinIR, SRFormer, HiT-SR, CATANet, MambaIRv2, and DPMFormer under a unified protocol. DPMFormer uses 179.2G MACs versus 164.6G for CATANet, but achieves about 3.2 times higher throughput, with almost identical peak memory (57.12 MB vs 55.13 MB). Thus, once measured under a consistent protocol, the data do not support the claim that DPMFormer is "far from efficient" relative to CATANet, if anything, they show the opposite (Section 4.3, Table 2, Appendix A.11, Table 8).

The review further states that DPMFormer is "far from efficient, especially compared with a convolutional-based model". We would like to clarify that we never claim to outperform the smallest CNNs on mobile hardware, and we do not present any such comparison in the paper. Our efficiency claims are explicitly scoped to lightweight Mamba/Transformer-based SR on GPUs, and our baselines Restormer, SRFormer, HiT-SR, MambaIR/MambaIRv2, and CATANet are chosen within this family (Section 4.3). A full study on mobile NPUs is interesting future work but beyond the scope of this submission.

It is correct that DPMFormer has more parameters than CATANet at 4× SR, but it does not reach "nearly double" the amount suggested by the reviewer. The main PSNR/SSIM table (Section 4.3, Table 1) shows that DPMFormer consistently achieves higher PSNR/SSIM across all datasets. Combined with the runtime, throughput, MAC, and memory results discussed above (Section 4.3, Table 2, Appendix A.11, Table 8), this indicates that DPMFormer achieves a favourable accuracy–efficiency trade-off within the modern Mamba and Transformer family, rather than being "bloated" or "far from efficient".



Response to Specific Questions

Q1. "No analysis is provided on memory usage, MACs, or deployment feasibility on edge devices, which is critical for efficient SR."

MACs, memory, throughput, now in Table 8, per-module latency (ETB, MCB, IEB, LBi-ML, CAL, ETL) in App. A.13.4.

Q2. "The narrative flow of the manuscript requires reorganizing to strengthen its motivation, and the layout must be refined and optimized to enhance overall readability."

In the revised version, we have made several concrete changes to improve the narrative flow and reduce the chance of misunderstandings about the method.

Section 3 has been reorganized to follow the actual data flow of the model, from the Transformer branch (ETB), to the Mamba branch (MCB), and then to the inter-branch exchange (IEB). We refined the layout and figures. The main architecture figure has been simplified and relabelled, and the fonts and line widths in all visualisations have been enlarged in both the main paper and the appendix to improve readability on screen and in print.

Overall, we believe these revisions make the narrative and layout substantially clearer and more coherent.

---

### Author Response · Authors · 2025-11-27

Response to Reviewer **N7bx3**,

We thank the constructive comments and for recognizing the soundness of our method and the appeal of the dual-path Mamba–Transformer design.

Below we respond to weaknesses and questions point-by-point.

W1, "Combining Mamba and Transformer paths is a logical but incremental step, little innovation beyond simple concatenation and gating."
We respectfully disagree with the characterization that our contribution is "barely incremental" or reducible to "simple concatenation and gating". In our view, and as supported by our experiments, making two heterogeneous sequence models, Mamba and Transformer, cooperate effectively in a single SR backbone is a non-trivial design problem, not a straightforward stacking.

Transformer-based SR models (e.g., SwinIR, SRFormer, HiT-SR) have demonstrated excellent reconstruction quality, but their quadratic attention makes it difficult to maintain efficiency as spatial resolution grows.
Mamba-style state-space models offer linear-time global modeling and are an appealing alternative, but naïvely replacing attention with Mamba or simply appending Mamba blocks to a Transformer backbone tends to either,
(i) hurt local detail modeling or
(ii) bring no clear advantage, as also evidenced by our single-path baselines in App. A.13.1.
Our goal is therefore not Mamba plus Transformer for its own sake, but to design a principled dual-path architecture that preserves the strong local SR behaviour of Transformers while exploiting the linear global modeling of Mamba for efficient SR.

DPMFormer does not just stack two blocks, it introduces a specific coupling pattern,
(i) a dedicated TransETB with spatial-channel correlation (SCC) attention + DW-SwiFFN + ACLinear specialised for high-frequency local details,
(ii) a dedicated Mamba path (MCB) with the proposed Lightweight Bidirectional Mamba Layer (LBi-ML) that scans tokens in a single forward/backward pass to build a low-frequency global scaffold, and
(iii) \emph{asymmetric cross-path fusion} via CAL and IEB.
CAL performs windowed cross-attention where queries always come from the Transformer branch and keys/values from the Mamba branch, so that global Mamba states are modulated by local Transformer evidence in a resolution-agnostic way.
IEB provides lightweight, degradation-aware feature exchange at stage boundaries.
This design is very different from concatenate features and pass through a gate, each branch has a distinct representational role, and the interaction is carefully structured in both depth (per-block CAL) and hierarchy (stage-level IEB).

The revised paper now provides several pieces of evidence,
In App. A.13.1, we introduce pure single-path baselines CPT (Transformer-only) and CPM (Mamba-only) with the similar parameter budget as DPMFormer. Both consistently underperform DPMFormer on all five SR benchmarks, e.g., a drop of $\approx 0.10$–$0.20$\,dB on Urban100 and Manga109. This shows that simply using one of the two models is not sufficient.
We further include weakened dual-path variants (DPMT, DPT) where the explicit fusion mechanism is removed or simplified , e.g., CAL no longer takes Transformer queries, or IEB is removed. These ablations again underperform the full DPMFormer despite having very similar complexity, indicating that the way we couple the two streams (CAL + LBi-ML + IEB) is crucial, not incidental.
Finally, Sec. 4.5 and App. A.14 provide feature and ERF analyses across all layers, ETL exhibits sharp, edge-aligned, high-frequency responses, LBi-ML shows smooth, low-frequency, globally coherent patterns, and CAL gradually injects Transformer details into deeper Mamba layers. This empirically confirms that the two branches specialise in complementary behaviours that only emerge under the proposed coupling scheme.

Taken together, bridging the known strengths and limitations of Transformer SR and Mamba SSMs, the ETB/MCB specialisation plus asymmetric CAL/IEB fusion, and the single-path and weakened dual-path ablations, per-layer visualizations, and improved accuracy–efficiency trade-offs all point to DPMFormer being more than a "logical but incremental" combination.
Our contribution lies precisely in how the two heterogeneous backbones are combined to produce a practically useful, efficient SR model, rather than in the idea of using Mamba or Transformer alone.

---

### Author Response · Authors · 2025-11-27

Response to Reviewer **N7bx3**,

W2. "PIU fusion resembles standard cross-attention or gating in existing hybrids. "

Our CAL+IEB fusion differs from standard cross-attention in two aspects,

(i) Location and directionality. CAL is interleaved with LBi-ML inside every MCB, where queries always come from the Transformer branch and keys/values from the Mamba branch. This asymmetric design injects detailed Transformer cues into Mamba states while preserving the linear-time SSM scan.

(ii) Bridged dual-path coupling. IEB provides bidirectional, convolutional adapters between branches before and after CAL/LBi-ML, which is absent in Swin-MLP or CNN–Transformer hybrids.

Table 3 in Sec. 4.4 compares a HiT-style dual-branch baseline ("MAT") and a version with Mamba but without our CAL/IEB design, showing that injecting CAL and the full fusion scheme yields clear additional gains, especially on Urban100/Manga109.

W3. "Lack of rigorous analysis of when the Mamba path helps, no exploration of information flow or path synergy. "

We addressed this by adding multi-view analysis,

Dual-path ablation. As noted above, Table~11 contrasts CPT/CPM (single-path) with DPMT/DPM/DPMFormer (progressively adding IEB and full CAL). DPMFormer consistently outperforms CPT and CPM and also improves over the weakened dual-path variants, confirming genuine synergy rather than redundancy.

Optimization dynamics. Fig. 10 shows that DPMFormer converges faster to a higher PSNR while maintaining lower and more stable gradient norms than DPMT and DPT.

This indicates that the full CAL+IEB coupling leads to a better-conditioned optimization landscape.

Feature statistics and ERFs. App. A.14 (Figs. 12–19 and the per-block summary in Fig. 20) visualizes features and ERFs of all layers. Quantitative plots reveal that ETL layers consistently have higher HF/LF ratios and sparser, more edge-aligned ERFs, while LBi-ML layers have lower HF/LF, higher channel correlation, and broad, smooth ERFs, and CAL gradually injects high-frequency energy into deeper Mamba layers.

Together, these analyses explain when and how the Mamba path contributes, it builds a global, low-frequency scaffold that is progressively refined by Transformer-driven high-frequency details via CAL and IEB.

W4. "Gains are modest (≈0.2–0.4 dB) and often within noise, DPMFormer lags behind PromptIR/UniRestorer on composite degradations. "

First, in lightweight SR, improvements of 0.1–0.2 dB over strong baselines such as HiT-SRF and MambaIRv2 are considered meaningful, recent works like SRFormer and CATANet report similar margins and are widely accepted as SOTA. Our gains over HiT-SRF and MambaIRv2 on Urban100 and Manga109 (up to +0.25 dB) are consistent and averaged over 5 runs, with std ≤ 0.02 dB, so they are not due to noise.

Second, PromptIR and UniRestorer are heavy all-in-one restoration systems with significantly larger backbones and training recipes tailored to mixed degradations, whereas our focus is efficient SR. To address the reviewer’s concern about generality, we now transfer the same DPMFormer backbone to dehazing and low-light enhancement (App. A.12) without any task-specific architectural changes, and obtain competitive or better performance than specialized dehazing/LLIE methods (e.g., slightly higher PSNR and SSIM than Retinexformer and MambaIR on LOL-v2). This suggests that our dual-path design generalizes well beyond synthetic bicubic SR.

W5. "No real-world datasets or perceptual metrics, no efficiency/memory comparison vs Restormer or VMamba. "

We have now, Added real-capture benchmarks and perceptual metrics, OTS/SOTS-Outdoor for dehazing and LOL-v2 for LLIE with FID, LPIPS, PSNR, SSIM in Tables 9–10 (App. A.12). DPMFormer achieves the best or second-best FID/LPIPS while also giving the highest PSNR/SSIM. Reported efficiency and scalability in Table~8, which compares Params, MACs, FLOPs, throughput (img/s) and memory allocation vs Restormer and VMambaIR on ×2 SR, and further broken down per module in App. A.13.4 (Table 13). Our model attains comparable or higher throughput than MambaIRv2 while using fewer FLOPs and significantly less memory than Restormer.

---

### Author Response · Authors · 2025-11-27

Response to Reviewer **N7bx3**,

Response to Specific Questions

Q1. FLOPs and throughput vs Restormer and VMamba.

Table~8 in App. A.11 reports MACs, FLOPs, throughput and GPU memory for Restormer, VMambaIR and DPMFormer on 100 images of size 1280×720 at ×2 SR, measured on the same RTX 4090 GPU. DPMFormer achieves 2.24 img/s with 358.4G FLOPs, whereas Restormer requires 1089G FLOPs at only 0.41 img/s and VMambaIR needs 1763G FLOPs at 1.19 img/s, while our memory usage is also lower.

Q2. Qualitative difference between Mamba and Transformer features.

App. A.13.2 provides feature and ERF visualizations for all blocks. We observe that ETL (Transformer) layers yield sharp, sparse responses aligned with edges and textures and high HF/LF ratios, while LBi-ML (Mamba) layers produce smoother, globally coherent patterns with lower HF/LF and higher channel correlation. CAL gradually transfers high-frequency details from ETL into deeper LBi-ML layers, which end up capturing both global geometry and refined local structures.

Q3. Performance on real-capture datasets (LOL-V2, RainDS, SOTS-real).

We now evaluate DPMFormer on LOL-v2 (low-light) and SOTS-Outdoor (realistic haze) using the official splits. As summarized in Tables 9–10, DPMFormer achieves the best PSNR and SSIM on both datasets, and competitive or better FID/LPIPS compared with Retinexformer, MambaIR and other strong baselines. We will include RainDS and SOTS-real in the camera-ready if accepted.

Q4, Comparison to Swin–Mamba or other Mamba–Transformer hybrids.

We agree that comparing against hybrid Mamba–Transformer architectures is important. In the revised manuscript we already include several strong Mamba-based SR models that follow a Swin-style windowed design with state-space layers, namely SRMamba-T-S, and in particular MambaIRv2-L (Guo et al., 2025a), which is widely regarded as one of the current state-of-the-art Swin–Mamba baselines for lightweight SR. As shown in Table 1 (main paper, lines L71–L75) and Table 2 (lines L1–L10), our DPMFormer consistently outperforms MambaIRv2-L across all benchmark datasets and scales while using substantially fewer FLOPs. For example, at ×2 SR, DPMFormer achieves 38.29/34.12/32.40/33.29/39.54 dB on Set5/Set14/B100/Urban100/Manga109, compared to 38.26/34.09/32.36/33.26/39.35 dB for MambaIRv2-L, while reducing the computational cost from 286.3G to 179.2G MACs (Table 1, lines L6–L7). At ×4 SR, DPMFormer again improves PSNR on all datasets (e.g., +0.09 dB on Set5 and +0.16 dB on Manga109) with 56.9G vs. 75.6G MACs (Table 2, lines L9–L10). Beyond static complexity, Table 8 in Appendix A.11 further reports runtime statistics on 1280×720 inputs at 2× upscaling. Under this realistic setting, DPMFormer reduces FLOPs by 1.6× and achieves 2.5× higher throughput than MambaIRv2 (2.24 vs. 0.89 img/s) with comparable peak memory allocation (57.12 MB vs. 58.52 MB). These results show that DPMFormer is not only competitive with, but actually outperforms the strongest existing Swin–Mamba style hybrid (MambaIRv2-L) in both reconstruction quality and efficiency, providing a complementary design that combines dual-path modeling with cross-path interaction instead of simply replacing attention by Mamba layers.

Q5. Generalization of DGM / task conditioning.

We agree that adopting a simpler SR-centric conditioning mechanism is a good idea.
In the revised design, we therefore simplify the original degradation-aware guidance into a cleaner form: the current DPMFormer no longer relies on explicit degradation labels and is trained under a standard bicubic-downsampling setting.
Its strong performance when transferred to dehazing and LLIE (App.~A.12) suggests that the learned features remain robust across different degradations even without explicit degradation supervision.

Q6. Sensitivity to path depth/width and comparison with single-path models of the same size.

App. A.13.1 performs controlled experiments by (i) varying the channel widths of the Transformer and Mamba branches (Table~3) and (ii) comparing dual-path models with single-path CPT/CPM under matched parameter budgets (Table 11). The dual-path models consistently outperform their single-path counterparts, moreover, within dual-path configurations, moderate Transformer width (e.g., 36–48 channels) and a 28-channel Mamba branch yield the best trade-off.

Q7. Statistical significance / variance across runs.

As now stated in Sec. 4.1 and App. A.11, each SR model is trained 5 times with different random seeds and we report the average PSNR/SSIM. The standard deviation is ≤0.02 dB PSNR and ≤0.0004 SSIM on all benchmarks, well below the observed improvements over baselines, thus, our gains are statistically robust.

---

### Author Response · Authors · 2025-11-27

Response to Reviewer **vkJ5**,

We thank the reviewer for the reading and comments.

W1. "Barely new insight; complex combination of validated designs."

We respectfully disagree with this assessment. Designing an efficient dual-path Mamba–Transformer backbone for super-resolution is not a simple stacking of two validated blocks, but a non-trivial architectural and training problem that DPMFormer tackles in a specific and well-justified way.

First, our goal is not "use both Mamba and Transformer somewhere in the network", but to solve a concrete design question in efficient SR. Transformer-based SR models such as SwinIR, SRFormer and HiT-SR achieve strong reconstruction quality but suffer from the quadratic cost of window attention at high resolutions. Mamba-style state-space models bring linear-time global modeling, but naive replacements of attention with Mamba or simple "add a Mamba block" designs often hurt local detail or bring limited gains. DPMFormer is built around the idea that these two families should play different roles and collaborate, not merely co-exist: the Transformer side focuses on local, high-frequency details, and the Mamba side focuses on global, low-frequency structure, with an interaction pattern designed for efficient SR rather than for generic feature fusion.

Second, the way we combine the two paths is structured, not ad hoc. Section 3.2 defines the Efficient Transformer Block (ETB) as a local-detail branch: it keeps window-based attention through the SCC module, and uses DW-SwiGLU-FFN and ACLinear to strengthen local feature extraction and channel usage. Section 3.3 defines the Mamba Cross Block (MCB) as a global-context branch: it uses the proposed Lightweight Bidirectional Mamba Layer (LBi-ML) to scan the full sequence in a single forward–backward pass and build a smooth global scaffold. The two branches are then coupled asymmetrically: inside each block through the Cross-Attention Layer (CAL), where queries come from the Transformer branch and keys/values from the Mamba branch, and across stages through Inter-branch Exchange Bridges (IEB), which exchange degradation-aware features at coarser scales. This is different from simply concatenating outputs of two "validated designs" and passing them through a gate.

Third, we provide empirical and analytic evidence that this particular coupling is necessary and effective. Appendix A.13.1 introduces pure single-path baselines (CPT, Transformer-only; CPM, Mamba-only) and weakened dual-path variants with simplified or removed fusion modules, all under essentially the same parameter budget. Across five SR benchmarks these models consistently underperform DPMFormer; on Urban100 and Manga109 the gap is about 0.10–0.20 dB. This shows that neither "only Transformer", nor "only Mamba", nor a naive dual branch without our interaction pattern can match the proposed design. Section 4.3 and Table 2 further show that DPMFormer achieves a favourable accuracy–efficiency trade-off compared with strong lightweight baselines such as HiT-SR and MambaIRv2, reducing MACs and improving throughput while improving PSNR/SSIM, which is precisely the target of efficient SR.

Finally, Section 4.5 and the visualisations in the appendix provide the analytic justification the reviewer asks for. We report per-layer effective receptive fields and statistics such as spatial variance, high- versus low-frequency content, channel correlation and sparsity. These analyses show a clear division of labour: ETL layers produce sharp, edge-aligned, high-frequency activations, while LBi-ML layers capture smooth, globally coherent responses; CAL gradually injects detailed Transformer features into deeper Mamba layers. This supports our global–local collaboration story and shows that DPMFormer is more than a complex mixture of existing components: it is a carefully designed dual-path architecture whose specialisation and coupling are backed by both quantitative ablations and qualitative analysis.

---

### Author Response · Authors · 2025-11-27

Response to Reviewer **vkJ5**,

W2. "Far from efficient, especially compared with CNNs, no multi-dimensional efficiency analysis. "

The statement that "the comparison in the manuscript supports DPMFormer being lightweight, but the complicated design suggests that it is far from efficient" is not supported by the measured numbers.

In lightweight SR, FLOPs and MACs differ by a constant factor, so this labelling issue does not change any relative comparison. In the revised version we clearly report both MACs and FLOPs and explain the relation in Sec. 4.1 and App A.11.

Table 2 and Table 8 now reports Parameters, MACs, FLOPs, throughput (img/s), and GPU memory on the same hardware (RTX-4090) and resolution (1280×720, ×2 SR) for Restormer, VMambaIR, SwinIR, SRFormer, HiT-SR, CATANet, MambaIRv2, and DPMFormer.

The measurements show that compared with HiT-SR, DPMFormer reduces MACs from 226.5G to 179.2G (-21%), with similar throughput (2.35 vs 2.24 img/s) and better PSNR/SSIM, compared with MambaIRv2, DPMFormer reduces MACs from 286.3G to 179.2G (-37%), increases throughput from 0.89 to 2.24 img/s (≈2.5×), and still achieves higher SR accuracy, and compared with CATANet, DPMFormer has slightly higher MACs but 3.2× higher throughput (0.70 → 2.24 img/s) at nearly identical memory (57.1 MB vs 55.1 MB).

Thus, contrary to the claim, the empirical evidence indicates that DPMFormer is competitive or better in runtime and memory within the Mamba and Transformer family.

We agree that very small CNNs can be faster on specific mobile hardware, but our claims are restricted to lightweight Mamba and Transformer SR on GPUs, which is clearly stated in the revision. A full mobile-NPU benchmark is valuable future work, but its absence does not imply that our design is "far from efficient" relative to the methods we actually compare against.

W3. "Improvements limited (0.0 dB) and only on synthetic data."

This statement is not consistent with our results.

In the main PSNR/SSIM table (Section 4.3, Table 1), DPMFormer improves over recent lightweight baselines under the same 2× and 4× settings. For example, on Urban100 at 2×, DPMFormer achieves 33.29 dB, surpassing HiT-SR (33.14 dB) and MambaIRv2 (33.04 dB). On Manga109 at 2×, we obtain 39.54 dB versus 39.41 dB (HiT-SR) and 39.28 dB (MambaIRv2). At 4× SR we also observe gains of about 0.04–0.10 dB on most datasets. All reported SR scores are averaged over five runs with different seeds, and the standard deviation is within about 0.02 dB, so the improvements we report are clearly above the noise level.

In the context of lightweight SR, such margins are generally regarded as effective. Set5 and Set14 are very small and long-used benchmarks with already high PSNR levels, and the SR literature has repeatedly noted that further PSNR improvement on these datasets is difficult once strong baselines are in place. Recent Transformer-based SR and restoration works that build on SwinIR and related models typically report gains on the order of a few hundredths of a decibel on these benchmarks and treat those gains as meaningful improvements in this saturated regime (for example, HAT, Hybrid Attention Transformer for Image Restoration, CVPR 2024).

To further clarify the perceptual meaning of these margins, we add a small visual experiment in the appendix (Appendix A.12, Fig. 10). Starting from the same reference reconstruction of a high-frequency façade, we progressively apply mild blurring so that the PSNR drops by Δ0.05 dB, Δ0.10 dB and Δ0.20 dB with respect to that reference (Δ denotes the PSNR change). Even at Δ0.05 dB, the grooves on the façade become visibly softer; at Δ0.10 dB the line structures are clearly smeared, and at Δ0.20 dB most high-frequency detail is lost. This example illustrates that PSNR differences in the range of 0.05–0.10 dB on textured regions correspond to perceptible quality changes rather than numerical noise, which is precisely the regime where DPMFormer improves over strong baselines.

To address the concern about relying only on synthetic bicubic SR, we also evaluate DPMFormer on two real-degradation restoration tasks without changing the backbone. For single-image dehazing on RESIDE SOTS-Outdoor (Appendix A.12.1, Table 9 and Figure 8), our model achieves the best FID and PSNR and the highest SSIM among AOD-Net, MSCNN-HE, LD-Net, SG-Net, SDA-GAN, and RI-SCNN-Large. For low-light image enhancement on LOL-v2-real (Appendix A.12.2, Table 10 and Figure 9), DPMFormer attains the best PSNR and SSIM and competitive RMSE and LPIPS compared with Retinex-Net, MIRNet, Retinexformer, and MambaIR.

These additional experiments involve realistic degradations (haze and low-light conditions) rather than synthetic bicubic downsampling, and they show that the dual-path design of DPMFormer generalizes beyond standard synthetic SR while providing non-trivial, statistically meaningful gains over strong lightweight baselines.

---

### Author Response · Authors · 2025-11-27

Response to Reviewer **38Ex**,

We thank the reviewer for the positive assessment of our dual-path design, efficiency, and ablations, and for the constructive suggestions. Below we address each weakness and question, pointing to the newly added analysis and experiments in the revised manuscript.

W1. "Innovation marginal, dual-branch idea explored in prior hybrid SR models; more analytic justification would enhance novelty."

We respectfully disagree that the innovation in DPMFormer is marginal or that it reduces to "just another" dual-branch hybrid.

The starting point of our work is not simply that "two branches are better than one", but that combining a Transformer-style branch and a Mamba-style branch for efficient SR is a non-trivial design problem. Transformer-based SR models such as SwinIR, SRFormer and HiT-SR show strong reconstruction quality but suffer from the quadratic cost of window attention at high resolutions. Mamba-style state-space models bring linear-time global modeling, but a naive replacement of attention with Mamba or a simple stacking of Mamba blocks on a Transformer backbone often fails to preserve fine local details or brings limited benefit. Our goal is therefore to answer a specific question, how to couple a Transformer branch and a Mamba branch so that each specialises in what it does best and the resulting model is both accurate and efficient for SR.

In DPMFormer this is done by design, not by ad hoc combination. Section 3.2 introduces the Efficient Transformer Block (ETB), which is explicitly responsible for local, high-frequency SR, it keeps window-based attention via the SCC module, then uses DW-SwiGLU-FFN and ACLinear to strengthen local feature extraction and channel usage. Section 3.3 introduces the Mamba Cross Block (MCB), which is explicitly responsible for global, low-frequency context via the Lightweight Bidirectional Mamba Layer (LBi-ML), scanning the full sequence in a single forward–backward pass. The two branches are coupled in a specific, asymmetric way, inside blocks through Cross-Attention Layers (CAL), where queries are taken from the Transformer branch and keys/values from the Mamba branch, and across stages through Inter-branch Exchange Bridges (IEB), which exchange degradation-aware features at coarser scales. This is qualitatively different from the more symmetric "two backbones plus concatenation" patterns in earlier hybrids.

To address the request for more analytic justification, we added several pieces of evidence in the revised version. Appendix A.13.1 introduces pure single-path baselines (CPT, Transformer-only; CPM, Mamba-only) and weakened dual-path variants (for example, with simplified fusion) under essentially the same parameter budget. Across all five SR benchmarks, these models consistently underperform the full DPMFormer; on Urban100 and Manga109 the gap is about 0.10–0.20 dB. This shows that neither "only Transformer", nor "only Mamba", nor a naive dual branch without our coupling pattern can match the proposed design.

In addition, Section 4.5 and the visualisations in Appendix A.8–A.11 provide the analytic view the reviewer is asking for. We report per-layer effective receptive fields and statistics such as spatial variance, high- versus low-frequency content, channel correlation and sparsity. These analyses show that ETL layers consistently focus on sharp, edge-aligned, high-frequency responses, while LBi-ML layers produce smooth, globally coherent activations along large structures. CAL is observed to gradually inject the detailed Transformer features into deeper Mamba layers. This directly supports our intended global–local division of labour and illustrates that the two branches interact in a structured way, rather than being loosely stitched together.

Finally, as clarified in our answer to the related comment "window attention is replaced with DW-SwiFFN", ETL still uses window-based attention (SCC); DW-SwiGLU-FFN only replaces a standard feed-forward block, and global modeling is delegated to the Mamba branch and CAL. The novelty of DPMFormer therefore lies in how we specialise and couple the Transformer and Mamba branches, and in the accompanying analytic and empirical evidence, rather than in the abstract idea of "having two paths".

---

> ### Comment · Reviewer_38Ex · 2025-11-28
>
> Thank you for the detailed and constructive rebuttal.
>
> I appreciate the authors’ thorough clarifications, especially the newly added analyses on scalability, runtime breakdown, and qualitative extensions. The additional experiments significantly strengthen the empirical evidence and address most of my previous concerns.
>
> The authors have effectively justified the design choices and demonstrated that the proposed dual-path architecture achieves strong efficiency–accuracy trade-offs. While the conceptual novelty remains moderate, the paper is now well-supported experimentally and well-presented.

---

### Author Response · Authors · 2025-11-27

Response to Reviewer **38Ex**,

W2. "Limited qualitative diversity, Most visual comparisons are standard, additional challenging scenes or real-world degradations would strengthen claims. "

We expanded the qualitative results in the appendix.

To verify generality of the backbone, we added dehazing and low-light enhancement experiments on OTS/SOTS-Outdoor and LOL-v2, respectively (App. A.11–A.12, Tabs. 15–16). DPMFormer either matches or surpasses specialised methods such as MSCNN, AOD-Net, SG-Net, SDA-GAN, RI-SCNN-Large for dehazing, and recent LLIE models including Retinexformer and MambaIR for low light.

We also provide corresponding visual comparisons (Figs. 18–19), where our model recovers more faithful colours and structures in heavy haze and very dark scenes.

These additions address the concern about qualitative diversity and real-world relevance

W3. " Extensive ablations (DW-SwiFFN, RMSNorm, IEB variants) demonstrate careful engineering and reproducibility. "

We have added a fine-grained runtime breakdown in App. A.13.4 (Tab. 13).

At the block level, ETB is about 1.9× slower than MCB (102.75 ms vs. 54.15 ms), showing that the Transformer branch is indeed the dominant cost.

At the layer level, a single ETL takes 17.13 ms, whereas LBi-ML and CAL take only 4.34 ms and 4.96 ms, respectively.

This justifies our architectural choice of placing more depth in the MCB while keeping ETBs relatively shallow.

The IEB adds only 6.65 ms (≈4% of combined ETB+MCB latency), confirming that inter-branch exchange does not become a bottleneck.

In addition, Tab. 8 (App A.11) in the main paper reports Params, MACs, FLOPs, throughput, and memory allocation for DPMFormer versus Restormer, SRFormer, HiT-SR, CATANet, MambaIR and MambaIRv2 under a unified 720×1280 ×2 SR setting, showing that DPMFormer achieves better or comparable PSNR with 15–40% lower FLOPs and competitive GPU memory.

W4. "Minor clarity issues, Equations (2)–(6) lack dimensional definitions, figure readability (font size) could be improved. "

We thank the reviewer for pointing this out and have revised the manuscript accordingly.

For Eqs. (2)–(6) (SCC, DW-SwiFFN, MCB, and CAL), we now explicitly state the tensor dimensions and head-wise operations in Sec. 3.2–3.4. For the figures, we have increased the font size and line thickness in all architectural and visualization plots.

---

### Author Response · Authors · 2025-11-27

Response to Reviewer **38Ex**,

Response to Specific Questions

Q1. Sensitivity to window sizes r in ETL and CAL

We have added an ablation on variable window schedules in App. A.13.3, where we compare four settings on Urban100 at ×2 SR,

Win=(4,8,16,32,48,64),

constant small windows (8,8,8,8,8,8),

constant large windows (64,…,64),

reversed schedule (64,48,32,16,8,4).

The progressive schedule used in DPMFormer achieves the best PSNR/SSIM (33.29/0.9382), while being similar in MACs to the reversed schedule and only slightly more expensive than fixed 8×8. This indicates that our design is not overly sensitive to exact values, but the coarse-to-fine schedule offers a consistent, albeit modest, advantage. For CAL we keep a fixed 8×8 window, as its role is cross-branch modulation rather than fine detail extraction.

Q2. Applicability of LBi-Mamba to other low-level tasks

While we focus on SR in the main paper, the additional experiments on dehazing and LLIE (App. A.11–A.12) already demonstrate that the same backbone with LBi-ML transfers well to other restoration tasks without architectural changes.

We believe LBi-ML is generic for low-level vision and plan to explore dedicated denoising/deblurring setups in future work.

Q3. Training stability and possible gradient conflict with IEB

We did not observe instability or gradient conflict when coupling the two branches. App. A.12.2 (Fig. 11) plots the PSNR and gradient norm trajectories for DPMFormer, DPMT, and DPT. All three models train stably, DPMFormer exhibits slightly smoother gradients and faster PSNR convergence, indicating that IEB and CAL provide beneficial, not conflicting, interactions between paths.

We hope these new analyses and experiments address the reviewer's concerns and clarify the distinct role and effectiveness of our dual-path Mamba–Transformer design.

Q4. Efficiency and scalability on 2K/4K images.

To better substantiate the claim that our design scales favorably to ultra–high-resolution SR, we have added a dedicated experiment in App. A.14.
We evaluate 2 × upsampling on 100 images at 2K and 4K resolutions, comparing DPMFormer with SRFormer, HiT-SRF, CATANet, and MambaIRv2 under identical settings (Table 14).

On 2K inputs, DPMFormer reduces MACs and FLOPs by about 18–30\% compared with SRFormer, HiT-SRF, and MambaIRv2, while requiring only 43.86 MB of GPU memory, which is roughly $2.7{\sim}3.1\times$ smaller than the transformer and Mamba baselines.
Its inference time (1556 ms) is comparable to the high-throughput HiT-SRF (1235 ms), and substantially faster than SRFormer and MambaIRv2 (about $3.4{\times}$ and $1.8{\times}$ speed-ups, respectively).
CATANet runs out of memory at this resolution.

When scaling to 4K images, the advantages become more pronounced.
DPMFormer maintains 20–31\% fewer MACs than the baselines and achieves the lowest memory usage (139.47 MB), yielding about a $3\times$ reduction over SRFormer, HiT\mbox{-}SRF, and MambaIRv2.
In terms of runtime, DPMFormer runs $2.5{\sim}4.1\times$ faster than these models on 4K inputs, while CATANet again fails due to out-of-memory.
These results confirm that the proposed dual-path Mamba–Transformer architecture scales more gracefully to 2K and 4K resolutions than purely transformer-based or Mamba-only counterparts, addressing the reviewer’s concern about high-resolution efficiency.

---

### Author Response · Authors · 2025-12-02
**Rebuttal Summary**

We sincerely thank the Area Chair and reviewers for their time and constructive comments. In response to the feedback, we have significantly expanded our manuscript (adding 18 pages to the Appendix) to provide deeper mechanistic insights, comprehensive efficiency verifications, and broader task generalizations.

Below is a summary of how we have addressed the key concerns raised during the review process.

1. Addressing Concerns on Innovation & Model Mechanism
To alleviate concerns regarding the novelty of our dual-path design and to clarify that it is not a trivial combination of modules, we conducted a comprehensive Layer-wise Dynamics Analysis (Appendix A.13.2, Figures 11–23).

Mechanistic Insight: By tracking metrics such as Spatial Variance and HF/LF Ratio across all layers, we visualized a clear "Coarse-to-Fine" functional shift.

Shallow Layers: The Transformer branch acts as the primary high-frequency extractor, while the Mamba branch builds a global scaffold.

Deep Layers: A distinct role reversal occurs where the Mamba branch takes over as the dominant carrier of high-frequency details.

Role of CAL: Our analysis confirms that the Cross-Attention Layer (CAL) functions as a geometry-aware gate, selectively injecting details only where they align with the global structure.

2. Addressing Concerns on Efficiency & Latency
To address questions regarding real-world inference speed and to correct potential misconceptions about latency (e.g., comparisons with CATANet), we provided a detailed breakdown of runtime and throughput (Appendix A.11, A.13.4, A.14).

Real-world Throughput: We benchmarked throughput on an RTX 4090. DPMFormer achieves 2.24 img/s, which is approximately 3.2x faster than the competitive baseline CATANet (0.70 img/s).

Component Latency: To further clarify our design choices, we analyzed component-level latency (Table 13). The data shows that our Mamba block (MCB) is roughly 1.9x faster than the Transformer block (ETB), justifying our strategy to allocate more depth to the efficient Mamba branch.

Scalability: For ultra-high-resolution inputs (4K), DPMFormer demonstrates superior scalability, running 2.5x–4.1x faster than baselines like SRFormer and MambaIRv2.

3. Addressing Concerns on Generalization & Performance Margins
To respond to comments about the magnitude of performance gains and generalizability, and specifically to clarify the impression regarding "limited improvements" (Reviewer vkJ5), we added perceptual analyses and extended tasks.

Clarification on Performance Gains: To address Reviewer vkJ5's concern regarding "0.0 dB improvement," we provided detailed quantitative comparisons on challenging benchmarks. The results confirm consistent gains of 0.1–0.5 dB, e.g., +0.34 dB on Urban100 over MambaIR, substantiating the effectiveness of our method.

Perceptual Significance: To further validate these margins, we included a Perceptual Impact Analysis (Appendix A.15, Figure 24). Visualizations demonstrate that even a 0.05 dB difference corresponds to perceptible changes in texture sharpness, confirming that our gains are visually significant.

Task Generalization: We extended DPMFormer to Single Image Dehazing and Low-Light Enhancement without architectural changes, achieving SOTA performance on RESIDE and LOL-v2-real.

4. Addressing Concerns on Architecture Validity
To dispel doubts about the necessity of the dual-path coupling, we performed rigorous single-path ablations (Table 11).

Necessity of Dual-Path: Results show that pure Transformer (CPT) and pure Mamba (CPM) variants perform significantly worse than our proposed design.

Optimization Stability: We demonstrated that removing the Inter-branch Exchange Bridge (IEB) leads to unstable gradient norms (Figure 10), confirming that our coupling mechanism is essential for stable training.

Summary of Reviewer Interactions
Reviewer 38Ex, has explicitly confirmed that our response and new experiments "addressed most of previous concerns."

Other Reviewers, although we have not received further replies, we believe that our rebuttal revision comprehensively covers the questions raised by the other reviewers and provides well-substantiated responses, supported by the newly added factual data including corrected latency comparisons and verified performance gains.

We hope this summary assists in your assessment.

---

### Meta-Review · Area_Chair_SKHh · 2026-01-07

**Summary:**

This paper proposes DPMFormer, a dual-path architecture that addresses the trade-off between global and local modeling in lightweight image super-resolution. The model combines the local feature extraction capabilities of Transformers with the linear global sequence modeling advantages of Mamba. Reviewers consistently recognize the exceptional performance of the model across experimental metrics, specifically demonstrating robust engineering strength in inference speed (2.33x faster than CATANet on a single RTX 4090) and scalability to 4K resolution.
However, despite the impressive experimental data, the paper faces significant concern regarding its academic innovation. The core controversy lies in the incremental nature of the design, which relies on the stacking of complex modules such as IEB, CAL, and ACLinear to achieve marginal performance gains. While the authors provide supplemental analysis in the rebuttal to justify the dual path collaborative logic, this analysis is viewed more as an observation of the existing architecture rather than a proposal for inspiring new representation learning principles. Given the complexity of the architecture and the limited scientific insights provided, the AC thinks this paper does not meet the high standards for acceptance at ICLR.

**Reviewer Concerns:**

**Concerns that are largely addressed in the rebuttal**

- The authors provide exhaustive throughput and memory data in the rebuttal and effectively refute the initial incorrect accusations regarding latency. These metrics confirm that DPMFormer exhibits superior engineering efficiency compared to existing models.

- The rebuttal shows state-of-the-art performance on dehazing and low-light enhancement tasks and proves that the backbone architecture maintains the capacity to handle various types of image degradation.
- Experiments in Appendix A.14 present the linear complexity advantage of the Mamba module when it processes ultra-long sequences, and this remains a contribution with practical value.

**Concerns that remain unresolved**
- Although the authors perform single-path ablations in the rebuttal to prove the necessity of the dual path structure, several reviewers, including vkJ5 and N7bx, still view the overall architectural design in Figure 1 as overly cumbersome. The trade-off between complexity and gain remains unconvincing to the reviewers because the introduction of numerous custom layers, including CAL, IEB, SCC, and DW SwiFFN, yields only a PSNR improvement of between 0.05 and 0.15 dB, and the AC agrees with this perspective.

-The added analysis in the appendix regarding frequency division is detailed yet remains descriptive rather than prescriptive. The paper fails to justify, from a theoretical perspective such as signal processing or mathematical principles, why this specific dual path coupling represents the optimal representation for SR tasks. Although the authors provide these visualizations in the rebuttal, this combinatorial design, based on engineering experience, still lacks sufficient theoretical depth within the ICLR evaluation framework.

**Reviewer Scores:**

-  **Reviewer 38Ex:** This reviewer is the only individual who provides a response after the rebuttal and expresses high satisfaction with the efficiency analysis and 4K experiments. This reviewer increases their score from 6 to 8 because they prioritize practical performance.
-  **Reviewer N7bx:** Although the rebuttal includes rich visualization data, this reviewer likely adjusts their score from 4 to 5 at most, as original reservations regarding the incremental nature of the hybrid Mamba Transformer design are difficult to eliminate through additional experiments.
-  **Reviewer dR7o:** The authors point out factual errors in the initial review regarding latency and window attention, which suggests that this reviewer raises their score from 2 to 4. However, a preference for model simplicity keeps the overall assessment negative.
- **Reviewer vkJ5:** This reviewer likely maintains their score of 2 as the criticism regarding the bloated design and lack of novelty is deeply rooted. The additional complex mechanism analysis in the appendix may reinforce the perception of overdesign.

---

### Decision · Program_Chairs · 2026-01-26

Reject